# PUZZLES: A Benchmark for Neural Algorithmic Reasoning

**Benjamin Estermann**
ETH Zurich
`estermann@ethz.ch`

**Luca A. Lanzendörfer**
ETH Zurich
`lanzendoerfer@ethz.ch`

**Yannick Niedermayr**
ETH Zurich
`yannickn@ethz.ch`

**Roger Wattenhofer**
ETH Zurich
`wattenhofer@ethz.ch`

## Abstract

Algorithmic reasoning is a fundamental cognitive ability that plays a pivotal role in problem-solving and decision-making processes. Reinforcement Learning (RL) has demonstrated remarkable proficiency in tasks such as motor control, handling perceptual input, and managing stochastic environments. These advancements have been enabled in part by the availability of benchmarks. In this work we introduce PUZZLES, a benchmark based on Simon Tatham's Portable Puzzle Collection, aimed at fostering progress in algorithmic and logical reasoning in RL. PUZZLES contains 40 diverse logic puzzles of adjustable sizes and varying levels of complexity; many puzzles also feature a diverse set of additional configuration parameters. The 40 puzzles provide detailed information on the strengths and generalization capabilities of RL agents. Furthermore, we evaluate various RL algorithms on PUZZLES, providing baseline comparisons and demonstrating the potential for future research. All the software, including the environment, is available at `https://github.com/ETH-DISCO/rlp`.

Human intelligence relies heavily on logical and algorithmic reasoning as integral components for solving complex tasks. While Machine Learning (ML) has achieved remarkable success in addressing many real-world challenges, logical and algorithmic reasoning remains an open research question [1–7]. This research question is supported by the availability of benchmarks, which allow for a standardized and broad evaluation framework to measure and encourage progress [8–10].

Reinforcement Learning (RL) has made remarkable progress in various domains, showcasing its capabilities in tasks such as game playing [11–15] , robotics [16–19] and control systems [20–22]. Various benchmarks have been proposed to enable progress in these areas [23–29]. More recently, advances have also been made in the direction of logical and algorithmic reasoning within RL [30–32]. Popular examples also include the games of Chess, Shogi, and Go [33, 34]. Given the importance of logical and algorithmic reasoning, we propose a benchmark to guide future developments in RL and more broadly machine learning.

Logic puzzles have long been a playful challenge for humans, and they are an ideal testing ground for evaluating the algorithmic and logical reasoning capabilities of RL agents. A diverse range of puzzles, similar to the Atari benchmark [24], favors methods that are broadly applicable. Unlike tasks with a fixed input size, logic puzzles can be solved iteratively once an algorithmic solution is found. This allows us to measure how well a solution attempt can adapt and generalize to larger inputs. Furthermore, in contrast to games such as Chess and Go, logic puzzles have a known solution, making reward design easier and enabling tracking progress and guidance with intermediate rewards.

38th Conference on Neural Information Processing Systems (NeurIPS 2024) Track on Datasets and Benchmarks.

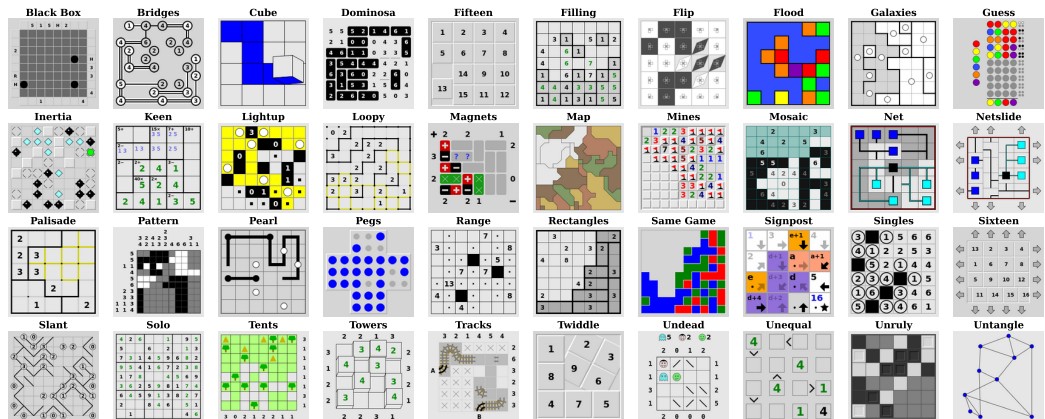

*Figure 1:* All puzzle classes of Simon Tatham's Portable Puzzle Collection.

In this paper, we introduce PUZZLES, a comprehensive RL benchmark specifically designed to evaluate RL agents' algorithmic reasoning and problem-solving abilities in the realm of logical and algorithmic reasoning. Simon Tatham's Puzzle Collection [35], curated by the renowned computer programmer and puzzle enthusiast Simon Tatham, serves as the foundation of PUZZLES. This collection includes a set of 40 logic puzzles, shown in Figure 1, each of which presents distinct challenges with various dimensions of adjustable complexity. They range from more well-known puzzles, such as *Solo* or *Mines* (commonly known as *Sudoku* and *Minesweeper*, respectively) to lesser-known puzzles such as *Cube* or *Slant*. PUZZLES includes all 40 puzzles in a standardized environment, each playable with a visual or discrete input and a discrete action space.

**Contributions.** We propose PUZZLES, an RL environment based on Simon Tatham's Puzzle Collection, comprising a collection of 40 diverse logic puzzles. To ensure compatibility, we have extended the original C source code to adhere to the standards of the Pygame library. Subsequently, we have integrated PUZZLES into the Gymnasium framework API, providing a straightforward, standardized, and widely-used interface for RL applications. PUZZLES allows the user to arbitrarily scale the size and difficulty of logic puzzles, providing detailed information on the strengths and generalization capabilities of RL agents. Furthermore, we have evaluated various RL algorithms on PUZZLES, providing baseline comparisons and demonstrating the potential for future research.

## 1 Related Work

**RL benchmarks.** Various benchmarks have been proposed in RL. Bellemare et al. [24] introduced the influential Atari-2600 benchmark, on which Mnih et al. [11] trained RL agents to play the games directly from pixel inputs. This benchmark demonstrated the potential of RL in complex, high-dimensional environments. PUZZLES allows the use of a similar approach where only pixel inputs are provided to the agent. Todorov et al. [23] presented MuJoCo which provides a diverse set of continuous control tasks based on a physics engine for robotic systems. Another control benchmark is the DeepMind Control Suite by Duan et al. [26], featuring continuous actions spaces and complex control problems. The work by Côté et al. [28] emphasized the importance of natural language understanding in RL and proposed a benchmark for evaluating RL methods in text-based domains. Lanctot et al. [29] introduced OpenSpiel, encompassing a wide range of games, enabling researchers to evaluate and compare RL algorithms' performance in game-playing scenarios. These benchmarks and frameworks have contributed significantly to the development and evaluation of RL algorithms. OpenAI Gym by Brockman et al. [25], and its successor Gymnasium by the Farama Foundation [36], helped by providing a standardized interface for many benchmarks. As such, Gym and Gymnasium

have played an important role in facilitating reproducibility and benchmarking in reinforcement learning research. Therefore, we provide PUZZLES as a Gymnasium environment to enable ease of use.

**Logical and algorithmic reasoning within RL.** Notable research in RL on logical reasoning includes automated theorem proving using deep RL [16] or RL-based logic synthesis [37]. Dasgupta et al. [38] find that RL agents can perform a certain degree of causal reasoning in a meta-reinforcement learning setting. The work by Jiang and Luo [30] introduces Neural Logic RL, which improves interpretability and generalization of learned policies. Eppe et al. [39] provide steps to advance problem-solving as part of hierarchical RL. Fawzi et al. [31] and Mankowitz et al. [32] demonstrate that RL can be used to discover novel and more efficient algorithms for well-known problems such as matrix multiplication and sorting. Neural algorithmic reasoning has also been used as a method to improve low-data performance in classical RL control environments [40, 41]. Logical reasoning might be required to compete in certain types of games such as chess, shogi and Go [33, 34, 42, 13], Poker [43–46] or board games [47–50]. However, these are usually multi-agent games, with some also featuring imperfect information and stochasticity.

**Reasoning benchmarks.** Various benchmarks have been introduced to assess different types of reasoning capabilities, although only in the realm of classical ML. IsarStep, proposed by Li et al. [8], specifically designed to evaluate high-level mathematical reasoning necessary for proof-writing tasks. Another significant benchmark in the field of reasoning is the CLRS Algorithmic Reasoning Benchmark, introduced by Veličković et al. [9]. This benchmark emphasizes the importance of algorithmic reasoning in machine learning research. It consists of 30 different types of algorithms sourced from the renowned textbook "Introduction to Algorithms" by Cormen et al. [51]. The CLRS benchmark serves as a means to evaluate models' understanding and proficiency in learning various algorithms. In the domain of large language models (LLMs), BIG-bench has been introduced by Srivastava et al. [10]. BIG-bench incorporates tasks that assess the reasoning capabilities of LLMs, including logical reasoning.

Despite these valuable contributions, a suitable and unified benchmark for evaluating logical and algorithmic reasoning abilities in single-agent perfect-information RL has yet to be established. Recognizing this gap, we propose PUZZLES as a relevant and necessary benchmark with the potential to drive advancements and provide a standardized evaluation platform for RL methods that enable agents to acquire algorithmic and logical reasoning abilities.

## 2 The PUZZLES Environment

In the following section, we give an overview of the PUZZLES environment.[1] The environment is written in both Python and C. For a detailed explanation of all features of the environment as well as their implementation, please see Appendices B and C.

### 2.1 Environment Overview

Within the PUZZLES environment, we encapsulate the tasks presented by each logic puzzle by defining consistent state, action, and observation spaces. It is also important to note that the large majority of the logic puzzles are designed so that they can be solved without requiring any guesswork. By default, we provide the option of two observation spaces, one is a representation of the discrete internal game state of the puzzle, the other is a visual representation of the game interface. These observation spaces can easily be wrapped in order to enable PUZZLES to be used with more advanced neural architectures such as graph neural networks (GNNs) or Transformers. All puzzles provide a discrete action space which only differs in cardinality. To accommodate the inherent difficulty and the need for proper algorithmic reasoning in solving these puzzles, the environment allows users to implement their own reward structures, facilitating the training of successful RL agents. All puzzles are played in a two-dimensional play area with deterministic state transitions, where a transition only occurs after a valid user input. Most of the puzzles in PUZZLES do not have an upper bound on the number of steps, they can only be completed by successfully solving the puzzle. An agent with a bad

---

[1] The puzzles are available to play online at `https://www.chiark.greenend.org.uk/~sgtatham/puzzles/`; excellent standalone apps for Android and iOS exist as well.

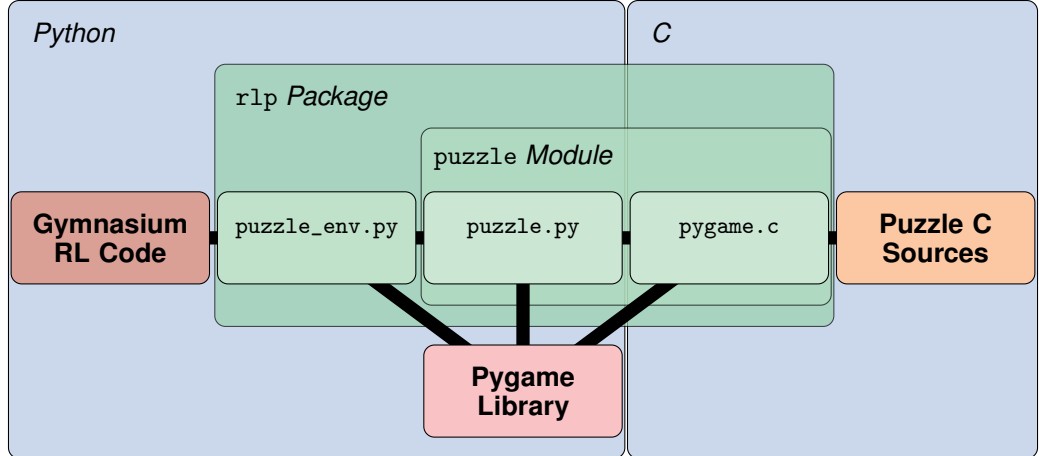

*Figure 2:* Code and library landscape around the PUZZLES Environment, made up of the rlp Package and the puzzle Module . The figure shows how the puzzle Module presented in this paper fits within Tathams's Puzzle Collection[2] code, the Pygame package, and a user's Gymnasium reinforcement learning code. The different parts are also categorized as Python language and C language.

policy is likely never going to reach a terminal state. For this reason, we provide the option for early episode termination based on state repetitions. As we show in Section 3.4, this is an effective method to facilitate learning.

## 2.2 Difficulty Progression and Generalization

The PUZZLES environment places a strong emphasis on giving users control over the difficulty exhibited by the environment. For each puzzle, the problem size and difficulty can be adjusted individually. The difficulty affects the complexity of strategies that an agent needs to learn to solve a puzzle. As an example, *Sudoku* has tangible difficulty options: harder difficulties may require the use of new strategies such as *forcing chains*[3] to find a solution, whereas easy difficulties only need the *single position* strategy.[4]

The scalability of the puzzles in our environment offers a unique opportunity to design increasingly complex puzzle configurations, presenting a challenging landscape for RL agents to navigate. This dynamic nature of the benchmark serves two important purposes. Firstly, the scalability of the puzzles facilitates the evaluation of an agent's generalization capabilities. In the PUZZLES environment, it is possible to train an agent in an easy puzzle setting and subsequently evaluate its performance in progressively harder puzzle configurations. For most puzzles, the cardinality of the action space is independent of puzzle size. It is therefore also possible to train an agent only on small instances of a puzzle and then evaluate it on larger sizes. This approach allows us to assess whether an agent has learned the correct underlying algorithm and generalizes to out-of-distribution scenarios. Secondly, it enables the benchmark to remain adaptable to the continuous advancements in RL methodologies. As RL algorithms evolve and become more capable, the puzzle configurations can be adjusted accordingly to maintain the desired level of difficulty. This ensures that the benchmark continues to effectively assess the capabilities of the latest RL methods.

## 3 Empirical Evaluation

We evaluate the baseline performance of numerous commonly used RL algorithms on our PUZZLES environment. Additionally, we also analyze the impact of certain design decisions of the environment and the training setup. Our metric of interest is the average number of steps required by a policy to

---

[3]*Forcing chains* works by following linked cells to evaluate possible candidates, usually starting with a two-candidate cell.

[4]The *single position* strategy involves identifying cells which have only a single possible value.

successfully complete a puzzle, where lower is better. We refer to the term *successful episode* to denote the successful completion of a single puzzle instance. We also look at the success rate, i.e. what percentage of the puzzles was completed successfully.

To provide an understanding of the puzzle's complexity and to contextualize the agents' performance, we include an upper-bound estimate of the optimal number of steps required to solve the puzzle correctly. This estimate is a combination of both the steps required to solve the puzzle using an optimal strategy, and an upper bound on the environment steps required to achieve this solution, such as moving the cursor to the correct position. The upper bound is denoted as *Optimal*. Please refer to Table 6 for details on how this upper bound is calculated for each puzzle. Further, we include the performance of a human expert as reference. The human expert is able to solve all puzzles in our evaluated difficulty levels within the optimal upper bound. For detailed results on the performance of the human expert, please refer to Appendix F.2.

We run experiments based on all the RL algorithms presented in Table 9. We include both popular traditional algorithms such as PPO, as well as algorithms designed more specifically for the kinds of tasks presented in PUZZLES. Where possible, we used the implementations available in the RL library Stable Baselines 3 [52], using the default hyperparameters. For MuZero and DreamerV3, we used the code available at [53] and [54], respectively. We provide a summary of all algorithms in Appendix Table 9. In total, our experiments required approximately 10'000 GPU hours.

All selected algorithms are compatible with the discrete action space required by our environment. This circumstance prohibits the use of certain other common RL algorithms, such as Soft-Actor Critic (SAC) [55] or Twin Delayed Deep Deterministic Policy Gradients (TD3) [56].

## 3.1 Baseline Experiments

For the general baseline experiments, we trained all agents on all puzzles and evaluate their performance. Due to the challenging nature of our puzzles, we have selected an easy difficulty and small size of the puzzle where possible. Every agent was trained on the discrete internal state observation using five different random seeds. We trained all agents by providing rewards only at the end of each episode upon successful completion or failure. For computational reasons, we truncated all episodes during training and testing at 10,000 steps. For such a termination, reward was kept at 0. We evaluate the effect of this episode truncation in Section 3.4. We provide all experimental parameters, including the exact parameters supplied for each puzzle in Appendix F.1.

To track an agent's progress, we use episode lengths, i.e., how many actions an agent needs to solve a puzzle. A lower number of actions indicates a stronger policy that is closer to the optimal solution. To obtain the final evaluation, we run each policy on 1000 random episodes of the respective puzzle, again with a maximum step size of 10,000 steps. All experiments were conducted on NVIDIA 3090 GPUs. The training time for a single agent with 2 million PPO steps varied depending on the puzzle and ranged from approximately 1.75 to 3 hours. The training for DreamerV3 and MuZero was more demanding and training time ranged from approximately 10 to 20 hours.

Figure 3b shows the average successful episode length for all algorithms, created following the recommendations outlined in [57]. It can be seen that DreamerV3 performs best when looking at success rate and episode length, with TRPO, PPO and DQN following closely. MuZero suffers from instable training, where a successful strategy was only learned for a low number of puzzles, indicating the need for puzzle-specific hyperparameter tuning. The results are especially interesting since PPO and TRPO follow much simpler training routines than DreamerV3 and MuZero. It seems that the implicit world models learned by DreamerV3 struggle to appropriately capture some puzzles. Upon closer inspection of the detailed results, presented in Appendix Table 10 and 11, DreamerV3 manages to solve 62.7% of all puzzle instances. In 14 out of the 40 puzzles, it has found a policy that solves the puzzles within the *Optimal* upper bound. PPO and TRPO managed to solve an average of 61.6% and 70.8% of the puzzle instances, however only 8 and 11 of the puzzles have consistently solved within the *Optimal* upper bound. The algorithms A2C, RecurrentPPO, DQN and QRDQN perform worse than a pure random policy. Overall, it seems that some of the environments in PUZZLES are quite challenging and well suited to show the difference in performance between algorithms. It is also important to note that all the logic puzzles are designed so that they can be solved without requiring any guesswork.

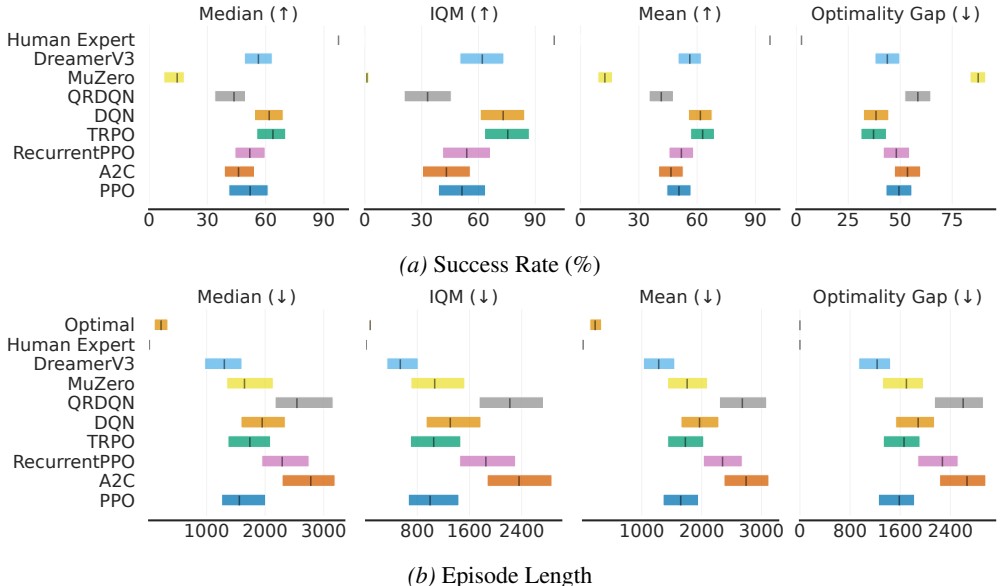

*(a) Success Rate (%)*

*(b) Episode Length*

*Figure 3:* Subfigure (a) shows the success rate aggregated over all puzzles in the easiest setting, while Subfigure (b) shows aggregated episode length. Interval estimates are based on stratified bootstrap confidence intervals, computed using `rliable` [57]. Some puzzles, namely Loopy, Pearl, Pegs, Solo, and Unruly, were intractable for all algorithms and were therefore excluded in this aggregation. Optimal refers to the upper bound of the performance of an optimal policy. A human expert is able to solve all puzzles within this bound. We report median, interquartile mean on the middle 50% of runs (IQM), mean, as well as the optimality gap with respect to the upper bound of an optimal policy. We see that DreamerV3, DQN and TRPO are able to solve the largest amount of puzzles, however, DreamerV3 seems to learn better policies. Overall, all algorithms fall short of optimal or human expert level performance.

## 3.2 Difficulty

We further evaluate the performance of a subset of the puzzles on the easiest preset difficulty level for humans. We selected all puzzles where a random policy was able to solve them with a probability of at least 10%, which are Netslide, Same Game and Untangle. By using this selection, we estimate that the reward density should be relatively high, ideally allowing the agent to learn a good policy. Again, we train all algorithms listed in Table 9. We provide results for the two strongest algorithms, PPO and DreamerV3 in Table 1, with complete results available in Appendix Table 10. Note that as part of Section 3.4, we also perform ablations using DreamerV3 on more puzzles on the easiest preset difficulty level for humans.

*Table 1:* Comparison of how many steps agents trained with PPO and DreamerV3 need on average to solve puzzles of two difficulty levels. In brackets, the percentage of successful episodes is reported. The difficulty levels correspond to the overall easiest and the easiest-for-humans settings. We also give the upper bound of optimal steps needed for each configuration.

| Puzzle | Parameters | PPO | DreamerV3 | Optimal | Human Expert |
|---|---|---|---|---|---|
| Netslide | 2x3b1 | $35.3 \pm 0.7$ (100.0%) | $12.0 \pm 0.4$ (100.0%) | 48 | 16.7 |
| | 3x3b1 | $4742.1 \pm 2960.1$ (9.2%) | $3586.5 \pm 676.9$ (22.4%) | 90 | 40.9 |
| Same Game | 2x3c3s2 | $11.5 \pm 0.1$ (100.0%) | $7.3 \pm 0.2$ (100.0%) | 42 | 8.7 |
| | 5x5c3s2 | $1009.3 \pm 1089.4$ (30.5%) | $527.0 \pm 162.0$ (30.2%) | 300 | 37.0 |
| Untangle | 4 | $34.9 \pm 10.8$ (100.0%) | $6.3 \pm 0.4$ (100.0%) | 80 | 6.0 |
| | 6 | $2294.7 \pm 2121.2$ (96.2%) | $1683.3 \pm 73.7$ (82.0%) | 150 | 30.5 |

We observe that for both PPO and DreamerV3, the percentage of successful episodes decreases, with a large increase in steps required. DreamerV3 performs clearly stronger than PPO, requiring

consistently fewer steps, but still more than the optimal policy. Our results indicate that puzzles with relatively high reward density at human difficulty levels remain challenging. We propose to use the easiest human difficulty level as a first measure to evaluate future algorithms. The details of the easiest human difficulty setting can be found in Appendix Table 7. If this level is achieved, difficulty can be further scaled up by increasing the size of the puzzles. Some puzzles also allow for an increase in difficulty with fixed size.

### 3.3 Effect of Action Masking and Observation Representation

We evaluate the effect of action masking, as well as observation type, on training performance. Firstly, we analyze whether action masking, as described in paragraph "Action Masking" in Appendix B.4, can positively affect training performance. Secondly, we want to see if agents are still capable of solving puzzles while relying on pixel observations. Pixel observations allow for the exact same input representation to be used for all puzzles, thus achieving a setting that is very similar to the Atari benchmark. We compare MaskablePPO to the default PPO without action masking on both types of observations. We summarize the results in Figure 4. Detailed results for masked RL agents on the pixel observations are provided in Appendix Table 12.

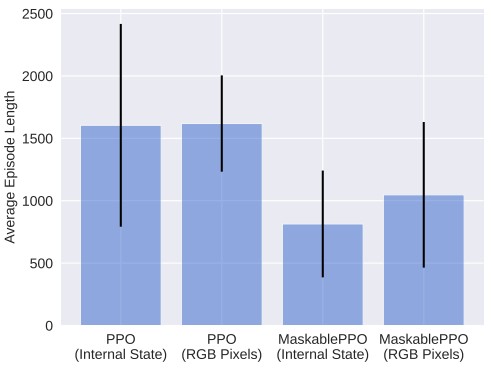
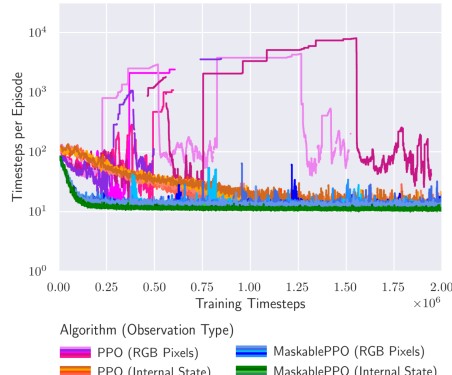

*Figure 4:* (left) We demonstrate the effect of action masking in both RGB observation and internal game state. By masking moves that do not change the current state, the agent requires fewer actions to explore, and therefore, on average solves a puzzle using fewer steps. (right) Moving average episode length during training for the *Flood* puzzle. Lower episode length is better, as the episode gets terminated as soon as the agent has solved a puzzle. Different colors describe different algorithms, where different shades of a color indicate different random seeds. Sparse dots indicate that an agent only occasionally managed to find a policy that solves a puzzle. It can be seen that both the use of discrete internal state observations and action masking have a positive effect on the training, leading to faster convergence and a stronger overall performance.

As we can observe in Figure 4, action masking has a strongly positive effect on training performance. This benefit is observed both in the discrete internal game state observations and on the pixel observations. We hypothesize that this is due to the more efficient exploration, as actions without effect are not allowed. As a result, the reward density during training is increased, and agents are able to learn a better policy. Particularly noteworthy are the outcomes related to *Pegs*. They show that an agent with action masking can effectively learn a successful policy, while a random policy without action masking consistently fails to solve any instance. As expected, training RL agents on pixel observations increases the difficulty of the task at hand. The agent must first understand how the pixel observation relates to the internal state of the game before it is able to solve the puzzle. Nevertheless, in combination with action masking, the agents manage to solve a large percentage of all puzzle instances, with 10 of the puzzles consistently solved within the optimal upper bound.

Furthermore, Figure 4 shows the individual training performance on the puzzle *Flood*. It can be seen that RL agents using action masking and the discrete internal game state observation converge significantly faster and to better policies compared to the baselines. The agents using pixel observations and no action masking struggle to converge to any reasonable policy.

## 3.4 Effect of Episode Length and Early Termination

We evaluate whether the cutoff episode length or early termination have an effect on training performance of the agents. For computational reasons, we perform these experiments on a selected subset of the puzzles on the easiest preset human level difficulty and only for DreamerV3 (see Appendix F.4 for details). As we can see in Table 2, increasing the maximum episode length during training from 10,000 to 100,000 does not improve performance. Only when episodes get terminated after visiting the exact same state more than 10 times, the agent is able to solve more puzzle instances on average (31.5% vs. 25.2%). Given the sparse reward structure, terminating episodes early seems to provide a better trade-off between allowing long trajectories to successfully complete and avoiding wasting resources on unsuccessful trajectories. Interestingly, the solution for the puzzle Cube found by DreamerV3 requires fewer steps than the human expert.

*Table 2:* Comparison of the effect of the maximum episode length (# Steps) and early termination (ET) on final performance. For each setting, we report average success episode length with standard deviation with respect to the random seed, all averaged over all selected puzzles. In brackets, the percentage of successful episodes is reported.

| #Steps | ET | DreamerV3 |
|---|---|---|
| $1e5$ | 10 | $2950.9 \pm 1260.2$ (31.6%) |
| | - | $2975.4 \pm 1503.5$ (25.2%) |
| $1e4$ | 10 | $3193.9 \pm 1044.2$ (26.1%) |
| | - | $2892.4 \pm 908.3$ (26.8%) |

## 3.5 Generalization

PUZZLES is explicitly designed to facilitate the testing of generalization capabilities of agents with respect to different puzzle sizes or puzzle difficulties. For our experiments, we select puzzles with the highest reward density. We utilize a custom observation wrapper and transformer-based encoder in order for the agent to be able to work with different input sizes, see Appendices A.3 and A.4 for details. We call this approach PPO (Transformer)

The results presented in Table 3 indicate that while it is possible to learn a policy that generalizes it remains a challenging problem. Furthermore, it can be observed that selecting the best model during training according to the performance on the generalization environment yields a performance benefit in that setting. This suggests that agents may learn a policy that generalizes better during the training process, but then overfit on the environment they are training on. It is also evident that generalization performance varies substantially across different random seeds. For Netslide, the best agent is capable of solving 23.3% of the puzzles in the generalization environment whereas the worst

*Table 3:* We test generalization capabilities of agents by evaluating them on puzzle sizes larger than their training environment. We report the average number of steps an agent needs to solve a puzzle, and the percentage of successful episodes in brackets. The difficulty levels correspond to the overall easiest and the easiest-for-humans settings. For PPO (Transformer), we selected the best checkpoint during training according to the performance in the training environment. For PPO (Transformer)[†], we selected the best checkpoint during training according to the performance in the generalization environment.

| Puzzle | Parameters | Trained on | PPO (Transformer) | PPO (Transformer)[†] |
|---|---|---|---|---|
| Netslide | 2x3b1 | ✓ | $244.1 \pm 313.7$ (100.0%) | $242.0 \pm 379.3$ (100.0%) |
| | 3x3b1 | ✗ | $9014.6 \pm 2410.6$ (18.6%) | $9002.8 \pm 2454.9$ (18.0%) |
| Same Game | 2x3c3s2 | ✓ | $9.3 \pm 10.9$ (99.8%) | $26.2 \pm 52.9$ (99.7%) |
| | 5x5c3s2 | ✗ | $379.0 \pm 261.6$ (9.4%) | $880.1 \pm 675.4$ (18.1%) |
| Untangle | 4 | ✓ | $38.6 \pm 58.2$ (99.8%) | $69.8 \pm 66.4$ (100.0%) |
| | 6 | ✗ | $3340.0 \pm 3101.2$ (87.3%) | $2985.8 \pm 2774.7$ (93.7%) |

agent is only able to solve 11.2% of the puzzles, similar to a random policy. Our findings suggest that agents are generally capable of generalizing to more complex puzzles. However, further research is necessary to identify the appropriate inductive biases that allow for consistent generalization without a significant decline in performance.

## 4 Discussion

The experimental evaluation demonstrates varying degrees of success among different algorithms. For instance, puzzles such as *Tracks*, *Map* or *Flip* were not solvable by any of the evaluated RL agents, or only with performance similar to a random policy. This points towards the potential of intermediate rewards, better game rule-specific action masking, or model-based approaches. To encourage exploration in the state space, a mechanism that explicitly promotes it may be beneficial. On the other hand, the fact that some algorithms managed to solve a substantial amount of puzzles with presumably optimal performance demonstrates the advances in the field of RL. In light of the promising results of DreamerV3, the improvement of agents that have certain reasoning capabilities and an implicit world model by design stay an important direction for future research.

**Experimental Results.** The experimental results presented in Section 3.1 and Section 3.3 underscore the positive impact of action masking and the correct observation type on performance. While a pixel representation would lead to a uniform observation for all puzzles, it currently increases complexity too much compared the discrete internal game state. Our findings indicate that incorporating action masking significantly improves the training efficiency of reinforcement learning algorithms. This enhancement was observed in both discrete internal game state observations and pixel observations. The mechanism for this improvement can be attributed to enhanced exploration, resulting in agents being able to learn more robust and effective policies. This was especially evident in puzzles where unmasked agents had considerable difficulty, thus showcasing the tangible advantages of implementing action masking for these puzzles.

**Limitations.** While the PUZZLES framework provides the ability to gain comprehensive insights into the performance of various RL algorithms on logic puzzles, it is crucial to recognize certain limitations when interpreting results. The sparse rewards used in this baseline evaluation add to the complexity of the task. Moreover, all algorithms were evaluated with their default hyperparameters. Additionally, the constraint of discrete action spaces excludes the application of certain RL algorithms.

**Benchmarking LLMs.** We also explore the potential of using PUZZLES as a novel framework for evaluating the reasoning abilities of both large language models (LLMs) and vision language models (VLMs). While existing reasoning benchmarks have been the subject of debate regarding their ability to accurately assess the reasoning abilities of LLMs [58–60], PUZZLES offers a unique advantage by enabling true out-of-distribution evaluation. Our preliminary experiments, conducted with Gemini 1.5 Flash [61] and GPT-4o mini [62], indicate that current LLMs have limited success in solving PUZZLES. A detailed analysis of these results is presented in Appendix F.5. This research lays the groundwork for future investigations using PUZZLES to provide a more nuanced understanding of the reasoning processes and limitations of LLMs.

In the context of RL, the different challenges posed by the logic-requiring nature of these puzzles necessitates a good reward system, strong guidance of agents, and an agent design more focused on logical reasoning capabilities. It will be interesting to see how alternative architectures such as graph neural networks (GNNs) perform. GNNs are designed to align more closely with the algorithmic solution of many puzzles. While the notion that "reward is enough" [63, 64] might hold true, our results indicate that not just *any* form of correct reward will suffice, and that advanced architectures might be necessary to learn an optimal solution.

## 5 Conclusion

In this work, we have proposed PUZZLES, a benchmark that bridges the gap between algorithmic reasoning and RL. In addition to containing a rich diversity of logic puzzles, PUZZLES also offers an adjustable difficulty progression for each puzzle, making it a useful tool for benchmarking, evaluating and improving RL algorithms. Our empirical evaluation shows that while RL algorithms exhibit

varying degrees of success, challenges persist, particularly in puzzles with higher complexity or those requiring nuanced logical reasoning. We are excited to share PUZZLES with the broader research community and hope that this benchmark will foster further research to improve the algorithmic reasoning abilities of machine learning models.

## Broader Impact

This paper aims to contribute to the advancement of the field of machine learning (ML). Given the current challenges in ML related to algorithmic reasoning, we believe that our newly proposed benchmark will facilitate significant progress in this area, potentially elevating the capabilities of ML systems. Progress in algorithmic reasoning can contribute to the development of more transparent, explainable, and fair ML systems. This can further help address issues related to bias and discrimination in automated decision-making processes, promoting fairness and accountability.

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

# A  PUZZLES Environment Usage Guide

## A.1  General Usage

A Python code example for using the PUZZLES environment is provided in Listing 1. All puzzles support seeding the initialization, by adding #{seed} after the parameters, where {seed} is an int. The allowed parameters are displayed in Table 6. A full custom initialization argument would be as follows: {parameters}#{seed}.

```python
import gymnasium as gym
import rlp

# init an agent suitable for Gymnasium environments
agent = Agent.create()

# init the environment
env = gym.make('rlp/Puzzle-v0', puzzle="bridges",
               render_mode="rgb_array", params="4x4#42")
observation, info = env.reset()

# complete an episode
terminated = False
while not terminated:
    action = agent.choose(env)  # the agent chooses the next action
    observation, reward, terminated, truncated, info = env.step(action)
env.close()
```

*Listing 1:* Code example of how to initialize an environment and have an agent complete one episode. The PUZZLES environment is designed to be compatible with the Gymnasium API. The choice of Agent is up to the user, it can be a trained agent or random policy.

## A.2  Custom Reward

A Python code example for implementing a custom reward system is provided in Listing 3. To this end, the environment's step() function provides the puzzle's internal state inside the info Python dict.

```python
import gymnasium as gym
class PuzzleRewardWrapper(gym.Wrapper):
    def step(self, action):
        obs, reward, terminated, truncated, info = self.env.step(action)
        # Modify the reward by using members of info["puzzle_state"]
        return obs, reward, terminated, truncated, info
```

*Listing 2:* Code example of a custom reward implementation using Gymnasium's Wrapper class. A user can use the game state information provided in info["puzzle_state"] to modify the rewards received by the agent after performing an action.

## A.3  Custom Observation

A Python code example for implementing a custom observation structure that is compatible with an agent using a transformer encoder. Here, we provide the example for Netslide, please refer to our GitHub for more examples.

```python
import gymnasium as gym
import numpy as np
class NetslideTransformerWrapper(gym.ObservationWrapper):
    def __init__(self, env):
        super(NetslideTransformerWrapper, self).__init__(env)
        self.original_space = env.observation_space

        self.max_length = 512
        self.embedding_dim = 16 + 4
        self.observation_space = gym.spaces.Box(
```

```
11            low=-1, high=1, shape=(self.max_length, self.embedding_dim,), dtype=np.float32
12        )
13
14        self.observation_space = gym.spaces.Dict(
15            {'obs': self.observation_space,
16             'len': gym.spaces.Box(low=0, high=self.max_length, shape=(1,),
17             dtype=np.int32)}
18        )
19
20    def observation(self, obs):
21        # The original observation is an ordereddict with the keys ['barriers', 'cursor_pos', 'height',
22        #  'last_move_col', 'last_move_dir', 'last_move_row', 'move_count', 'movetarget', 'tiles', 'width', 'wrapping']
23        # We are only interested in 'barriers', 'tiles', 'cursor_pos', 'height' and 'width'
24        barriers = obs['barriers']
25        # each element of barriers is an uint16, signifying different elements
26        barriers = np.unpackbits(barriers.view(np.uint8)).reshape(-1, 16)
27        # add some positional embedding to the barriers
28        embedded_barriers = np.concatenate(
29            [barriers, self.pos_embedding(np.arange(barriers.shape[0]), obs['width'], obs['height'])], axis=1)
30
31        tiles = obs['tiles']
32        # each element of tiles is an uint16, signifying different elements
33        tiles = np.unpackbits(tiles.view(np.uint8)).reshape(-1, 16)
34        # add some positional embedding to the tiles
35        embedded_tiles = np.concatenate(
36            [tiles, self.pos_embedding(np.arange(tiles.shape[0]), obs['width'], obs['height'])], axis=1)
37        cursor_pos = obs['cursor_pos']
38
39        embedded_cursor_pos = np.concatenate(
40            [np.ones((1, 16)), self.pos_embedding_cursor(cursor_pos, obs['width'], obs['height'])], axis=1)
41
42        embedded_obs = np.concatenate([embedded_barriers, embedded_tiles, embedded_cursor_pos], axis=0)
43
44        current_length = embedded_obs.shape[0]
45        # pad with zeros to accomodate different sizes
46        if current_length < self.max_length:
47            embedded_obs = np.concatenate(
48                [embedded_obs, np.zeros((self.max_length - current_length, self.embedding_dim))], axis=0)
49        return {'obs': embedded_obs, 'len': np.array([current_length])}
50
51    @staticmethod
52    def pos_embedding(pos, width, height):
53        # pos is an array of integers from 0 to width*height
54        # width and height are integers
55        # return a 2D array with the positional embedding, using sin and cos
56        x, y = pos % width, pos // width
57        # x and y are integers from 0 to width-1 and height-1
58        pos_embed = np.zeros((len(pos), 4))
59        pos_embed[:, 0] = np.sin(2 * np.pi * x / width)
60        pos_embed[:, 1] = np.cos(2 * np.pi * x / width)
61        pos_embed[:, 2] = np.sin(2 * np.pi * y / height)
62        pos_embed[:, 3] = np.cos(2 * np.pi * y / height)
63        return pos_embed
64
65    @staticmethod
66    def pos_embedding_cursor(pos, width, height):
67        # cursor pos goes from -1 to width or height
68        x, y = pos
69        x += 1
70        y += 1
71        width += 1
72        height += 1
73        pos_embed = np.zeros((1, 4))
74        pos_embed[0, 0] = np.sin(2 * np.pi * x / width)
75        pos_embed[0, 1] = np.cos(2 * np.pi * x / width)
76        pos_embed[0, 2] = np.sin(2 * np.pi * y / height)
77        pos_embed[0, 3] = np.cos(2 * np.pi * y / height)
78        return pos_embed
79
80
```

*Listing 3:* Code example of a custom observation implementation using Gymnasium's `Wrapper` class. A user can use the all elements of rpovided in the `obs` dict to create a custom observation. In this code example, the resulting observation is suitable for a transformer-based encoder.

## A.4   Generalization Example

In Listing 4, we show how a transformer-based features extractor can be built for Stable Baseline 3's PPO MultiInputPolicy. Together with the observations from Listing 2, this feature extractor can work with variable-length inputs. This allows for easy evaluation in environments of different sizes than the environment the agent was originally trained in.

```
1 import gymnasium as gym
2 import numpy as np
3 from stable_baselines3.common.torch_layers import BaseFeaturesExtractor
4 from stable_baselines3 import PPO
```

```
5  import torch
6  import torch.nn as nn
7  from torch.nn import TransformerEncoder, TransformerEncoderLayer
8
9  class TransformerFeaturesExtractor(BaseFeaturesExtractor):
10     def __init__(self, observation_space, data_dim, embedding_dim, nhead, num_layers, dim_feedforward, dropout
          =0.1):
11         super(TransformerFeaturesExtractor, self).__init__(observation_space, embedding_dim)
12         self.transformer = Transformer(embedding_dim=embedding_dim,
13                                        data_dim=data_dim,
14                                        nhead=nhead,
15                                        num_layers=num_layers,
16                                        dim_feedforward=dim_feedforward,
17                                        dropout=dropout)
18
19     def forward(self, observations: gym.spaces.Dict) -> torch.Tensor:
20         # Extract the 'obs' key from the dict
21         obs = observations['obs']
22         length = observations['len']
23         # all elements of length should be the same (we can't train on different puzzle sizes at the same time)
24         length = int(length[0])
25         obs = obs[:, :length]
26         # Return the embedding of the cursor token (which is last)
27         return self.transformer(obs)[:, -1, :]
28
29
30 class Transformer(nn.Module):
31     def __init__(self, embedding_dim, data_dim, nhead, num_layers, dim_feedforward, dropout=0.1):
32         super(Transformer, self).__init__()
33         self.embedding_dim = embedding_dim
34         self.data_dim = data_dim
35
36         self.lin = nn.Linear(data_dim, embedding_dim)
37
38         encoder_layers = TransformerEncoderLayer(
39             d_model=self.embedding_dim,
40             nhead=nhead,
41             dim_feedforward=dim_feedforward,
42             dropout=dropout,
43             batch_first=True
44         )
45
46         self.transformer_encoder = TransformerEncoder(encoder_layers, num_layers)
47
48     def forward(self, x):
49         # x is of shape (batch_size, seq_length, embedding_dim)
50         x = self.lin(x)
51         transformed = self.transformer_encoder(x)
52         return transformed
53
54 if __name__ == "__main__":
55     policy_kwargs = dict(
56         features_extractor_class=TransformerFeaturesExtractor,
57         features_extractor_kwargs=dict(embedding_dim=args.transformer_embedding_dim,
58                                        nhead=args.transformer_nhead,
59                                        num_layers=args.transformer_layers,
60                                        dim_feedforward=args.transformer_ff_dim,
61                                        dropout=args.transformer_dropout,
62                                        data_dim=data_dims[args.puzzle])
63     )
64
65     model = PPO("MultiInputPolicy",
66                 env,
67                 policy_kwargs=policy_kwargs,
68                 )
69
```

*Listing 4:* Code example of a transformer-based feature extractor written in PyTorch, compatible with Stable Baselines 3's PPO. This encoder design allows for variable-length inputs, enabling generalization to previously unseen puzzle sizes.

# B  Environment Features

## B.1  Episode Definition

An episode is played with the intention of solving a given puzzle. The episode begins with a newly generated puzzle and terminates in one of two states. To achieve a reward, the puzzle is either solved completely or the agent has failed irreversibly. The latter state is unlikely to occur, as only a few games, for example pegs or minesweeper, are able to terminate in a failed state. Alternatively, the episode can be terminated early. Starting a new episode generates a new puzzle of the same kind, with the same parameters such as size or grid type. However, if the random seed is not fixed, the puzzle is likely to have a different layout from the puzzle in the previous episode.

## B.2   Observation Space

There are two kinds of observations which can be used by the agent. The first observation type is a representation of the discrete internal game state of the puzzle, consisting of a combination of arrays and scalars. This observation is provided by the underlying code of Tathams's puzzle collection. The composition and shape of the internal game state is different for each puzzle, which, in turn, requires the agent architecture to be adapted.

The second type of observation is a representation of the pixel screen, given as an integer matrix of shape (3×width×height). The environment deals with different aspect ratios by adding padding. The advantage of the pixel representation is a consistent representation for all puzzles, similar to the Atari RL Benchmark [11]. It could even allow for a single agent to be trained on different puzzles. On the other hand, it forces the agent to learn to solve the puzzles only based on the visual representation of the puzzles, analogous to human players. This might increase difficulty as the agent has to learn the task representation implicitly.

## B.3   Action Space

Natively, the puzzles support two types of input, mouse and keyboard. Agents in PUZZLES play the puzzles only through keyboard input. This is due to our decision to provide the discrete internal game state of the puzzle as an observation, for which mouse input would not be useful.

The action space for each puzzle is restricted to actions that can actively contribute to changing the logical state of a puzzle. This excludes "memory aides" such as markers that signify the absence of a certain connection in *Bridges* or adding candidate digits in cells in *Sudoku*. The action space also includes possibly rule-breaking actions, as long as the game can represent the effect of the action correctly.

The largest action space has a cardinality of 14, but most puzzles only have five to six valid actions which the agent can choose from. Generally, an action is in one of two categories: selector movement or game state change. Selector movement is a mechanism that allows the agent to select game objects during play. This includes for example grid cells, edges, or screen regions. The selector can be moved to the next object by four discrete directional inputs and as such represents an alternative to continuous mouse input. A game state change action ideally follows a selector movement action. The game state change action will then be applied to the selected object. The environment responds by updating the game state, for example by entering a digit or inserting a grid edge at the current selector position.

## B.4   Action Masking

The fixed-size action space allows an agent to execute actions that may not result in any change in game state. For example, the action of moving the selector to the right if the selector is already placed at the right border. The PUZZLES environment provides an action mask that marks all actions that change the state of the game. Such an action mask can be used to improve performance of model-based and even some model-free RL approaches. The action masking provided by PUZZLES does not ensure adherence to game rules, rule-breaking actions can most often still be represented as a change in the game state.

## B.5   Reward Structure

In the default implementation, the agent only receives a reward for completing an episode. Rewards consist of a fixed positive value for successful completion and a fixed negative value otherwise. This reward structure encourages an agent to solve a given puzzle in the least amount of steps possible. The PUZZLES environment provides the option to define intermediate rewards tailored to specific puzzles, which could help improve training progress. This could be, for example, a negative reward if the agent breaks the rules of the game, or a positive reward if the agent correctly achieves a part of the final solution.

## B.6 Early Episode Termination

Most of the puzzles in PUZZLES do not have an upper bound on the number of steps, where the only natural end can be reached via successfully solving the puzzle. The PUZZLES environment also provides the option for early episode termination based on state repetitions. If an agent reaches the exact same game state multiple times, the episode can be terminated in order to prevent wasteful continuation of episodes that no longer contribute to learning or are bound to fail.

## C  PUZZLES Implementation Details

In the following, a brief overview of PUZZLES's code implementation is given. The environment is written in both Python and C, in order to interface with Gymnasium [36] as the RL toolkit and the C source code of the original puzzle collection. The original puzzle collection source code is available under the MIT License.[5] In maintext Figure 2, an overview of the environment and how it fits with external libraries is presented. The modular design in both PUZZLES and the Puzzle Collection's original code allows users to build and integrate new puzzles into the environment.

**Environment Class**   The reinforcement learning environment is implemented in the Python class `PuzzleEnv` in the `rlp` package. It is designed to be compatible with the Gymnasium-style API for RL environments to facilitate easy adoption. As such, it provides the two important functions needed for progressing an environment, `reset()` and `step()`.

Upon initializing a `PuzzleEnv`, a 2D surface displaying the environment is created. This surface and all changes to it are handled by the Pygame [65] graphics library. PUZZLES uses various functions provided in the library, such as shape drawing, or partial surface saving and loading.

The `reset()` function changes the environment state to the beginning of a new episode, usually by generating a new puzzle with the given parameters. An agent solving the puzzle is also reset to a new state. `reset()` also returns two variables, `observation` and `info`, where `observation` is a Python `dict` containing a NumPy 3D array called `pixels` of size (3 × surface_width × surface_height). This NumPy array contains the RGB pixel data of the Pygame surface, as explained in Appendix B.2. The `info` dict contains a `dict` called `puzzle_state`, representing a copy of the current internal data structures containing the logical game state, allowing the user to create custom rewards.

The `step()` function increments the time in the environment by one step, while performing an action chosen from the action space. Upon returning, `step()` provides the user with five variables, listed in Table 4.

*Table 4:* Return values of the environment's `step()` function. This information can then be used by an RL framework to train an agent.

| Variable | Description |
| --- | --- |
| observation | 3D NumPy array containing RGB pixel data |
| reward | The cumulative reward gained throughout all steps of the episode |
| terminated | A `bool` stating whether an episode was completed by the agent |
| truncated | A `bool` stating whether an episode was ended early, for example by reaching the maximum allowed steps for an episode |
| info | A `dict` containing a copy of the internal game state |

**Intermediate Rewards**   The environment encourages the use of Gymnasium's `Wrapper` interface to implement custom reward structures for a given puzzle. Such custom reward structures can provide an easier game setting, compared to the sparse reward only provided when finishing a puzzle.

**Puzzle Module**   The `PuzzleEnv` object creates an instance of the class `Puzzle`. A `Puzzle` is essentially the glue between all Pygame surface tasks and the C back-end that contains the puzzle

---

[5]The source code and license are available at `https://www.chiark.greenend.org.uk/~sgtatham/puzzles/`.

logic. To this end, it initializes a Pygame window, on which shapes and text are drawn. The `Puzzle` instance also loads the previously compiled shared library containing the C back-end code for the relevant puzzle.

The `PuzzleEnv` also converts and forwards keyboard inputs (which are for example given by an RL agent's action) into the format the C back-end understands.

**Compiled C Code**    The C part of the environment sits on top of the highly-optimized original puzzle collection source code as a custom front-end, as detailed in the collection's developer documentation [66]. Similar to other front-end types, it represents the bridge between the graphics library that is used to display the puzzles and the game logic back-end. Specifically, this is done using Python API calls to Pygame's drawing facilities.

## D  Puzzle Descriptions

We provide short descriptions of each puzzle from www.chiark.greenend.org.uk/ sgtatham/puzzles/. For detailed instructions for each puzzle, please visit the docs available at www.chiark.greenend.org.uk/ sgtatham/puzzles/doc/index.html

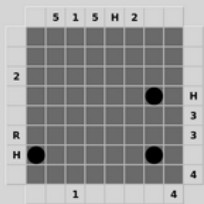

*Figure 5:* **Black Box**: Find the hidden balls in the box by bouncing laser beams off them.

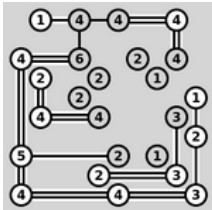

*Figure 6:* **Bridges**: Connect all the islands with a network of bridges.

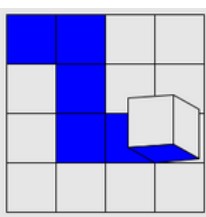

*Figure 7:* **Cube**: Pick up all the blue squares by rolling the cube over them.

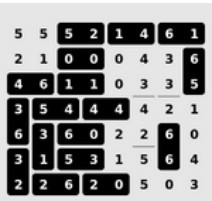

*Figure 8:* **Dominosa**: Tile the rectangle with a full set of dominoes.

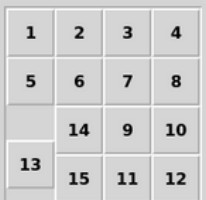

*Figure 9:* **Fifteen**: Slide the tiles around to arrange them into order.

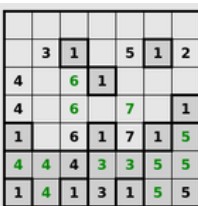

*Figure 10:* **Filling**: Mark every square with the area of its containing region.

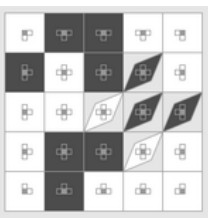

*Figure 11:* **Flip**: Flip groups of squares to light them all up at once.

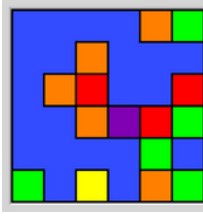

*Figure 12:* **Flood**: Turn the grid the same colour in as few flood fills as possible.

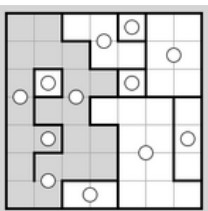

*Figure 13:* **Galaxies**: Divide the grid into rotationally symmetric regions each centred on a dot.

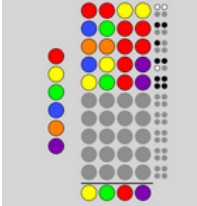

*Figure 14:* **Guess**: Guess the hidden combination of colours.

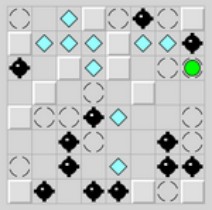

*Figure 15:* **Inertia**: Collect all the gems without running into any of the mines.

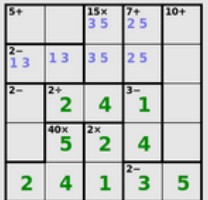

*Figure 16:* **Keen**: Complete the latin square in accordance with the arithmetic clues.

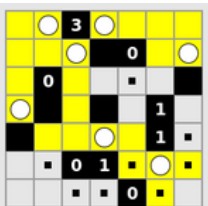

*Figure 17:* **Light Up**: Place bulbs to light up all the squares.

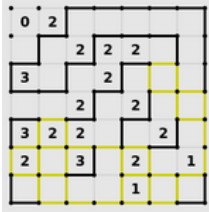

*Figure 18:* **Loopy**: Draw a single closed loop, given clues about number of adjacent edges.

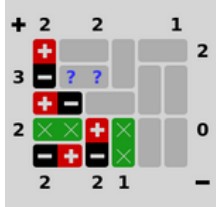

*Figure 19:* **Magnets**: Place magnets to satisfy the clues and avoid like poles touching.

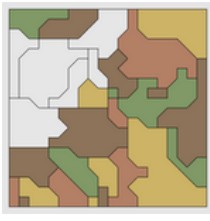

*Figure 20:* **Map**: Colour the map so that adjacent regions are never the same colour.

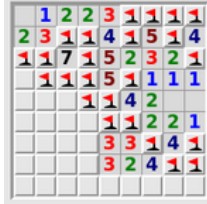

*Figure 21:* **Mines**: Find all the mines without treading on any of them.

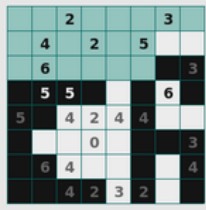

*Figure 22:* **Mosaic**: Fill in the grid given clues about number of nearby black squares.

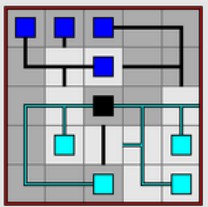

*Figure 23:* **Net**: Rotate each tile to reassemble the network.

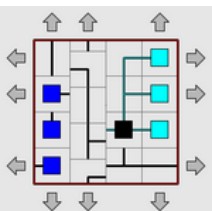

*Figure 24:* **Netslide**: Slide a row at a time to reassemble the network.

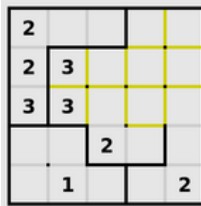

*Figure 25:* **Palisade**: Divide the grid into equal-sized areas in accordance with the clues.

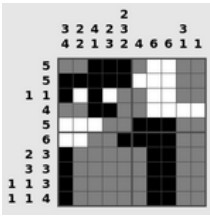

*Figure 26:* **Pattern**: Fill in the pattern in the grid, given only the lengths of runs of black squares.

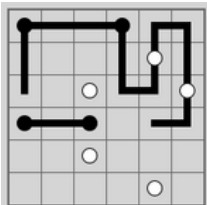

*Figure 27:* **Pearl**: Draw a single closed loop, given clues about corner and straight squares.

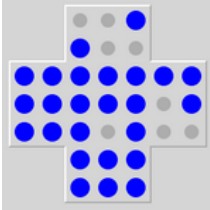

*Figure 28:* **Pegs**: Jump pegs over each other to remove all but one.

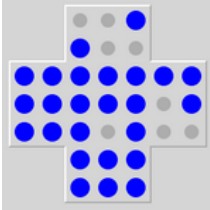

*Figure 29:* **Range**: Place black squares to limit the visible distance from each numbered cell.

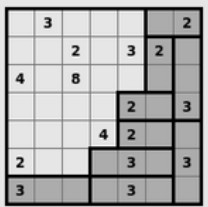

*Figure 30:* **Rectangles**: Divide the grid into rectangles with areas equal to the numbers.

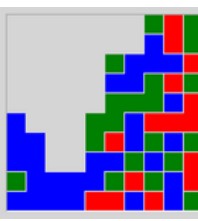

*Figure 31:* **Same Game**: Clear the grid by removing touching groups of the same colour squares.

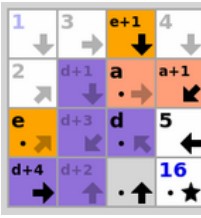

*Figure 32:* **Signpost**: Connect the squares into a path following the arrows.

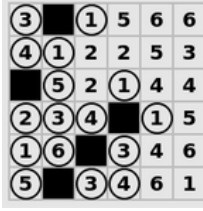

*Figure 33:* **Singles**: Black out the right set of duplicate numbers.

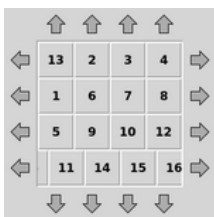

*Figure 34:* **Sixteen**: Slide a row at a time to arrange the tiles into order.

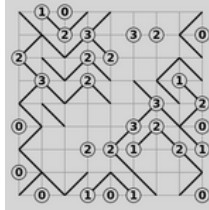

*Figure 35:* **Slant**: Draw a maze of slanting lines that matches the clues.

*Figure 36:* **Solo**: Fill in the grid so that each row, column and square block contains one of every digit.

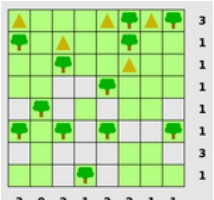

*Figure 37:* **Tents**: Place a tent next to each tree.

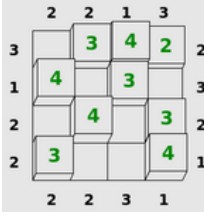

*Figure 38:* **Towers**: Complete the latin square of towers in accordance with the clues.

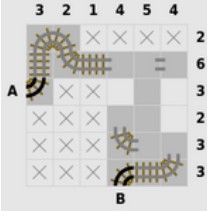

*Figure 39:* **Tracks**: Fill in the railway track according to the clues.

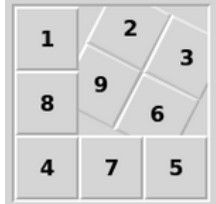

*Figure 40:* **Twiddle**: Rotate the tiles around themselves to arrange them into order.

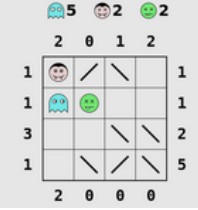

*Figure 41:* **Undead**: Place ghosts, vampires and zombies so that the right numbers of them can be seen in mirrors.

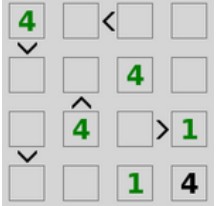

*Figure 42:* **Unequal**: Complete the latin square in accordance with the > signs.

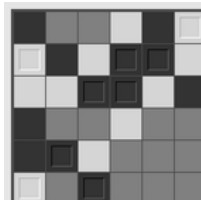

*Figure 43:* **Unruly**: Fill in the black and white grid to avoid runs of three.

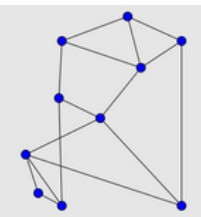 *Figure 44:* **Untangle**: Reposition the points so that the lines do not cross.

# E   Puzzle-specific Metadata

## E.1   Action Space

We display the action spaces for all supported puzzles in Table 5. The action spaces vary in size and in the types of actions they contain. As a result, an agent must learn the meaning of each action independently for each puzzle.

*Table 5:* The action spaces for each puzzle are listed, along with their cardinalities. The actions are listed with their name in the original Puzzle Collection C code.

| Puzzle | Cardinality | Action space |
|---|---|---|
| Black Box | 5 | UP, DOWN, LEFT, RIGHT, SELECT |
| Bridges | 5 | UP, DOWN, LEFT, RIGHT, SELECT |
| Cube | 4 | UP, DOWN, LEFT, RIGHT |
| Dominosa | 5 | UP, DOWN, LEFT, RIGHT, SELECT |
| Fifteen | 4 | UP, DOWN, LEFT, RIGHT |
| Filling | 13 | UP, DOWN, LEFT, RIGHT, 1, 2, 3, 4, 5, 6, 7, 8, 9 |
| Flip | 5 | UP, DOWN, LEFT, RIGHT, SELECT |
| Flood | 5 | UP, DOWN, LEFT, RIGHT, SELECT |
| Galaxies | 5 | UP, DOWN, LEFT, RIGHT, SELECT |
| Guess | 5 | UP, DOWN, LEFT, RIGHT, SELECT |
| Inertia | 9 | 1, 2, 3, 4, 6, 7, 8, 9, UNDO |
| Keen | 14 | UP, DOWN, LEFT, RIGHT, SELECT2, 1, 2, 3, 4, 5, 6, 7, 8, 9 |
| Light Up | 5 | UP, DOWN, LEFT, RIGHT, SELECT |
| Loopy | 6 | UP, DOWN, LEFT, RIGHT, SELECT, SELECT2 |
| Magnets | 6 | UP, DOWN, LEFT, RIGHT, SELECT, SELECT2 |
| Map | 5 | UP, DOWN, LEFT, RIGHT, SELECT |
| Mines | 7 | UP, DOWN, LEFT, RIGHT, SELECT, SELECT2, UNDO |
| Mosaic | 6 | UP, DOWN, LEFT, RIGHT, SELECT, SELECT2 |
| Net | 5 | UP, DOWN, LEFT, RIGHT, SELECT |
| Netslide | 5 | UP, DOWN, LEFT, RIGHT, SELECT |
| Palisade | 5 | UP, DOWN, LEFT, RIGHT, CTRL |
| Pattern | 6 | UP, DOWN, LEFT, RIGHT, SELECT, SELECT2 |
| Pearl | 5 | UP, DOWN, LEFT, RIGHT, SELECT |
| Pegs | 6 | UP, DOWN, LEFT, RIGHT, SELECT, UNDO |
| Range | 5 | UP, DOWN, LEFT, RIGHT, SELECT |
| Rectangles | 5 | UP, DOWN, LEFT, RIGHT, SELECT |
| Same Game | 6 | UP, DOWN, LEFT, RIGHT, SELECT, UNDO |
| Signpost | 6 | UP, DOWN, LEFT, RIGHT, SELECT, SELECT2 |
| Singles | 6 | UP, DOWN, LEFT, RIGHT, SELECT, SELECT2 |
| Sixteen | 6 | UP, DOWN, LEFT, RIGHT, SELECT, SELECT2 |
| Slant | 6 | UP, DOWN, LEFT, RIGHT, SELECT, SELECT2 |
| Solo | 13 | UP, DOWN, LEFT, RIGHT, 1, 2, 3, 4, 5, 6, 7, 8, 9 |
| Tents | 6 | UP, DOWN, LEFT, RIGHT, SELECT, SELECT2 |
| Towers | 14 | UP, DOWN, LEFT, RIGHT, SELECT2, 1, 2, 3, 4, 5, 6, 7, 8, 9 |
| Tracks | 5 | UP, DOWN, LEFT, RIGHT, SELECT |
| Twiddle | 6 | UP, DOWN, LEFT, RIGHT, SELECT, SELECT2 |
| Undead | 8 | UP, DOWN, LEFT, RIGHT, SELECT2, 1, 2, 3 |
| Unequal | 13 | UP, DOWN, LEFT, RIGHT, 1, 2, 3, 4, 5, 6, 7, 8, 9 |
| Unruly | 6 | UP, DOWN, LEFT, RIGHT, SELECT, SELECT2 |
| Untangle | 5 | UP, DOWN, LEFT, RIGHT, SELECT |

## E.2  Optional Parameters

We display the optional parameters for all supported puzzles in Table 6. If none are supplied upon initialization, a set of default parameters gets used for the puzzle generation process.

*Table 6:* For each puzzle, all optional parameters a user may supply are shown and described. We also give the required data type of variable, where applicable (e.g., int or char). For parameters that accept one of a few choices (such as difficulty), the accepted values and corresponding explanation are given in braces. As as example: a difficulty parameter is listed as d{int} with allowed values {0 = easy, 1 = medium, 2 = hard}. In this case, choosing medium difficulty would correspond to d1.

| Puzzle | Example | Parameter | Description | Optimal Step Upper Bound |
|---|---|---|---|---|
| Black Box | w8h8m5M5 | w{int} | grid width | $(w \cdot h + w + h + 1)$ |
| | | h{int} | grid height | $\cdot (w + 2) \cdot (h + 2)$ |
| | | m{int} | minimum number of balls | |
| | | M{int} | maximum number of balls | |
| Bridges | 7x7i5e2m2d0 | {int}x{int} | grid width $\times$ grid height | $3 \cdot w \cdot h \cdot (w + h + 8)$ |
| | | i{int} | percentage of island squares | |
| | | e{int} | expansion factor | |
| | | m{int} | max bridges per direction | |
| | | d{int} | difficulty {0 = easy, 1 = medium, 2 = hard} | |
| Cube | c4x4 | {char} | type {c = cube, t = tetrahedron, o = octahedron, i = icosahedron} | $w \cdot h \cdot F$ F = number of the body's faces |
| | | {int}x{int} | grid width $\times$ grid height | |
| Dominosa | 6db | {int} | maximum number of dominoes | $\frac{1}{2}\left(w^2 + 3w + 2\right)$ $\cdot (4\sqrt{w^2 + 3w + 2} + 1)$ |
| | | d{char} | difficulty {t = trivial, b = basic, h = hard, e = extreme, a = ambiguous} | |
| Fifteen | 4x4 | {int}x{int} | grid width $\times$ grid height | $(w \cdot h)^4$ |
| Filling | 13x9 | {int}x{int} | grid width $\times$ grid height | $(w \cdot h) \cdot (w + h + 1)$ |
| Flip | 5x5c | {int}x{int} | grid width $\times$ grid height | $(w \cdot h) \cdot (w + h + 1)$ |
| | | {char} | type {c = crosses, r = random} | |
| Flood | 12x12c6m5 | {int}x{int} | grid width $\times$ grid height | $(w \cdot h) \cdot (w + h + 1)$ |
| | | c{int} | number of colors | |
| | | m{int} | extra moves permitted (above the solver's minimum) | |
| Galaxies | 7x7dn | {int}x{int} | grid width $\times$ grid height | $(2 \cdot w \cdot h - w - h)$ $\cdot (2 \cdot w + 2 \cdot h + 1)$ |
| | | d{char} | difficulty {n = normal, u = unreasonable} | |
| Guess | c6p4g10Bm | c{int} | number of colors | $(p + 1) \cdot g \cdot (c + p)$ |
| | | p{int} | pegs per guess | |
| | | g{int} | maximum number of guesses | |
| | | {char} | allow blanks {B = no, b = yes} | |
| | | {char} | allow duplicates {M = no, m = yes} | |
| Inertia | 10x8 | {int}x{int} | grid width $\times$ grid height | $0.2 \cdot w^2 \cdot h^2$ |
| Keen | 6dn | {int} | grid size | $(2 \cdot w + 1) \cdot w^2$ |
| | | d{char} | difficulty {e = easy, n = normal, h = hard, x = extreme, u = unreasonable} | |
| | | {char} | (Optional) multiplication only {m = yes} | |
| Light Up | 7x7b20s4d0 | {int}x{int} | grid width $\times$ grid height | $\frac{1}{2} \cdot (w + h + 1)$ $\cdot (w \cdot h + 1)$ |
| | | b{int} | percentage of black squares | |
| | | s{int} | symmetry {0 = none, 1 = 2-way mirror, 2 = 2-way rotational, 3 = 4-way mirror, 4 = 4-way rotational} | |
| | | d{int} | difficulty {0 = easy, 1 = tricky, 2 = hard} | |
| Loopy | 10x10t12dh | {int}x{int} | grid width $\times$ grid height | $(2 \cdot w \cdot h + 1) \cdot 3 \cdot (w \cdot h)^2$ |
| | | t{int} | type {0 = squares, 1 = triangular, 2 = honeycomb, 3 = snub-square, 4 = cairo, 5 = great-hexagonal, 6 = octagonal, 7 = kites, 8 = floret, 9 = dodecagonal, 10 = great-dodecagonal, 11 = Penrose (kite/dart), 12 = Penrose (rhombs), 13 = great-great-dodecagonal, 14 = kagome, 15 = compass-dodecagonal, 16 = hats} | |
| | | d{char} | difficulty {e = easy, n = normal, t = tricky, h = hard} | |
| Magnets | 6x5dtS | {int}x{int} | grid width $\times$ grid height | $w \cdot h \cdot (w + h + 2)$ |
| | | d{char} | difficulty {e = easy, t = tricky | |
| | | {char} | (Optional) strip clues {S = yes} | |
| Map | 20x15n30dn | {int}x{int} | grid width $\times$ grid height | $2 \cdot n \cdot (1 + w + h)$ |
| | | n{int} | number of regions | |

Continued on next page

| Puzzle | Example | Parameter | Description | Optimal Step Upper Bound |
|--------|---------|-----------|-------------|--------------------------|
| | | d{char} | difficulty {e = easy, n = normal, h = hard, u = unreasonable} | |
| Mines | 9x9n10 | {int}x{int}
n{int}
p{char} | grid width $\times$ grid height
number of mines
(Optional) ensure solubility {a = no} | $w \cdot h \cdot (w + h + 1)$ |
| Mosaic | 10x10h0 | {int}x{int}
{str} | grid width $\times$ grid height
(Optional) aggressive generation {h0 = no} | $w \cdot h \cdot (w + h + 1)$ |
| Net | 5x5wb0.5 | {int}x{int}
{char}
b{float}
{char} | grid width $\times$ grid height
(Optional) walls wrap around {w = yes}
barrier probability, interval: [0, 1]
(Optional) ensure unique solution {a = no} | $w \cdot h \cdot (w + h + 3)$ |
| Netslide | 4x4wb1m2 | {int}x{int}
{char}
b{float}
m{int} | grid width $\times$ grid height
(Optional) walls wrap around {w = yes}
barrier probability, interval: [0, 1]
(Optional) number of shuffling moves | $2 \cdot w \cdot h \cdot (w + h - 1)$ |
| Palisade | 5x5n5 | {int}x{int}
n{int} | grid width $\times$ grid height
region size | $(2 \cdot w \cdot h - w - h)$
$\cdot (w + h + 3)$ |
| Pattern | 15x15 | {int}x{int} | grid width $\times$ grid height | $w \cdot h(w + h + 1)$ |
| Pearl | 8x8dtn | {int}x{int}
d{char}
{char} | grid width $\times$ grid height
difficulty {e = easy, t = tricky}
allow unsoluble {n = yes} | $w \cdot h \cdot (w + h + 2)$ |
| Pegs | 7x7cross | {int}x{int}
{str} | grid width $\times$ grid height
type {cross, octagon, random} | $w \cdot h \cdot (w + h + 2)$ |
| Range | 9x6 | {int}x{int} | grid width $\times$ grid height | $w \cdot h \cdot (w + h + 1)$ |
| Rectangles | 7x7e4 | {int}x{int}
e{int}
{char} | grid width $\times$ grid height
expansion factor
ensure unique solution {a = no} | $2 \cdot w \cdot h \cdot (w + h + 1)$ |
| Same Game | 5x5c3s2 | {int}x{int}
c{int}
s{int}

{char} | grid width $\times$ grid height
number of colors
scoring system {$1 = (n - 1)^2$,
$2 = (n - 2)^2$}
(Optional) ensure solubility {r = no} | $w \cdot h \cdot (w + h + 2)$ |
| Signpost | 4x4c | {int}x{int}
{char} | grid width $\times$ grid height
(Optional) start and end in corners
{c = yes} | $2 \cdot w \cdot h \cdot (w + h + 1)$ |
| Singles | 5x5de | {int}x{int}
d{char} | grid width $\times$ grid height
difficulty {e = easy, k = tricky} | $w \cdot h \cdot (w + h + 1)$ |
| Sixteen | 5x5m2 | {int}x{int}
m{int} | grid width $\times$ grid height
(Optional) number of shuffling moves | $w \cdot h \cdot (w + h + 3)$ |
| Slant | 8x8de | {int}x{int}
d{char} | grid width $\times$ grid height
difficulty {e = easy, h = hard} | $w \cdot h \cdot (w + h + 1)$ |
| Solo | 3x3 | {int}x{int}
{char}

{char}

{char}
{str}





d{char} | rows of sub-blocks $\times$ cols of sub-blocks
(Optional) require every digit on each
main diagonal {x = yes}
(Optional) jigsaw (irregularly shaped
sub-blocks) main diagonal {j = yes}
(Optional) killer (digit sums) {k = yes}
(Optional) symmetry. If not set,
it is 2-way rotation. {a = None,
m2 = 2-way mirror, m4 = 4-way mirror,
r4 = 4-way rotation, m8 = 8-way mirror,
md2 = 2-way diagonal mirror,
md4 = 4-way diagonal mirror}
difficulty {t = trivial, b = basic,
i = intermediate, a = advanced,
e = extreme, u = unreasonable} | $(w \cdot h)^2 * (2 \cdot w \cdot h + 1)$ |
| Tents | 8x8de | {int}x{int}
d{char} | grid width $\times$ grid height
difficulty {e = easy, t = tricky} | $\frac{1}{4} \cdot (w + 1) \cdot (h + 1)$
$\cdot (w + h + 1)$ |
| Towers | 5de | {int}
d{char} | grid size
difficulty {e = easy, h = hard
x = extreme, u = unreasonable} | $2 \cdot (w + 1) \cdot w^2$ |

| Puzzle | Example | Parameter | Description | Optimal Step Upper Bound |
|---|---|---|---|---|
| Tracks | 8x8dto | {int}x{int}
d{char}
{char} | grid width $\times$ grid height
difficulty {e = easy, t = tricky, h = hard}
(Optional) disallow consecutive 1 clues
{o = no} | $w \cdot h(2 \cdot (w + h) + 1)$ |
| Twiddle | 3x3n2 | {int}x{int}
n{int}
{char}
{char}
m{int} | grid width $\times$ grid height
rotating block size
(Optional) one number per row {r = yes}
(Optional) orientation matters {o = yes}
(Optional) number of shuffling moves | $(2 \cdot w \cdot h \cdot n^2 + 1)$
$\cdot (w + h - 2 \cdot n + 1)$ |
| Undead | 4x4dn | {int}x{int}
d{char} | grid width $\times$ grid height
difficulty {e = easy, n = normal, t = tricky} | $w \cdot h \cdot (w + h + 1)$ |
| Unequal | 4adk | {int}
{char}
d{char} | grid size
(Optional) adjacent mode {a = yes}
difficulty {t = trivial, e = easy, k = tricky,
x = extreme, r = recursive} | $w^2 \cdot (2 \cdot w + 1)$ |
| Unruly | 8x8dt | {int}
{char}
d{char} | grid size
(Optional) unique rows and cols {u = yes}
difficulty {t = trivial, e = easy, n = normal} | $w \cdot h \cdot (w + h + 1)$ |
| Untangle | 25 | {int} | number of points | $n \cdot (n + \sqrt{3n} \cdot 4 + 2)$ |

# F Detailed Results

## F.1 Baseline Parameters

In Table 7, the parameters used for training the agents used for the comparisons in Section 3 is shown.

*Table 7:* Listed below are the generation parameters supplied to each puzzle instance before training an agent, as well as some puzzle-specific notes. We propose the easiest preset difficulty setting as a first challenge for RL algorithms to reach human-level performance.

| Puzzle | Supplied Parameters | Easiest Human Level Preset | Notes |
|---|---|---|---|
| Black Box | w2h2m2M2 | w5h5m3M3 | |
| Bridges | 3x3 | 7x7i30e10m2d0 | |
| Cube | c3x3 | c4x4 | |
| Dominosa | 1dt | 3dt | |
| Fifteen | 2x2 | 4x4 | |
| Filling | 2x3 | 9x7 | |
| Flip | 3x3c | 3x3c | |
| Flood | 3x3c6m5 | 12x12c6m5 | |
| Galaxies | 3x3de | 7x7dn | |
| Guess | c2p3g10Bm | c6p4g10Bm | Episodes were terminated and negatively rewarded after the maximum number of guesses was made without finding the correct solution. |
| Inertia | 4x4 | 10x8 | |
| Keen | 3dem | 4de | Even the minimum allowed problem size proved to be infeasible for a random agent |
| Light Up | 3x3b20s0d0 | 7x7b20s4d0 | |
| Loopy | 3x3t0de | 7x7t0de | |
| Magnets | 3x3deS | 6x5de | |
| Map | 3x3n5de | 20x15n30de | |
| Mines | 4x4n2 | 9x9n10 | |
| Mosaic | 3x3 | 3x3 | |
| Net | 2x2 | 5x5 | |
| Netslide | 2x3b1 | 3x3b1 | |
| Palisade | 2x3n3 | 5x5n5 | |
| Pattern | 3x2 | 10x10 | |
| Pearl | 5x5de | 6x6de | |
| Pegs | 4x4random | 5x7cross | |
| Range | 3x3 | 9x6 | |
| Rectangles | 3x2 | 7x7 | |
| Same Game | 2x3c3s2 | 5x5c3s2 | |
| Signpost | 2x3 | 4x4c | |
| Singles | 2x3de | 5x5de | |
| Sixteen | 2x3 | 3x3 | |
| Slant | 2x2de | 5x5de | |
| Solo | 2x2 | 2x2 | |
| Tents | 4x4de | 8x8de | |
| Towers | 3de | 4de | |
| Tracks | 4x4de | 8x8de | |
| Twiddle | 2x3n2 | 3x3n2r | |
| Undead | 3x3de | 4x4de | |
| Unequal | 3de | 4de | |
| Unruly | 6x6dt | 8x8dt | Even the minimum allowed problem size proved to be infeasible for a random agent |
| Untangle | 4 | 6 | |

## F.2 Human Expert Evaluation

In order to provide more context on the difficulty of the puzzles, we report the results of a human expert solving all puzzles using the difficulties defined in Table 7.

*Table 8:* Number of steps required on average to solve the puzzles by a human expert. The human expert was able to solve 100% of the puzzles.

| Puzzle | Config | Episode Length | Puzzle | Config | Episode Length |
|---|---|---|---|---|---|
| Black Box | w2h2m2M2 | 20.3 | Palisade | 2x3n3 | 17.0 |
| | w5h5m3M3 | 120.0 | | 5x5n5 | 147.0 |
| Bridges | 3x3 | 8.3 | Pattern | 3x2 | 15.3 |
| | 7x7m2 | 49.3 | | 10x10 | 660.0 |
| Cube | c3x3 | 46.0 | Pearl | 5x5de | 46.0 |
| | c4x4 | 31.0 | | 6x6de | 245.0 |
| Dominosa | 1dt | 10.7 | Pegs | 4x4random | 27.3 |
| | 3dt | 51.5 | | 5x7cross | 253.0 |
| Fifteen | 2x2 | 4.7 | Range | 3x3 | 7.7 |
| | 4x4 | 287.0 | | 9x6 | 305.5 |
| Filling | 2x3 | 8.0 | Rect | 3x2 | 7.0 |
| | 9x7 | 123.0 | | 7x7 | 168.5 |
| Flip | 3x3c | 20.3 | Samegame | 2x3c3s2 | 8.7 |
| | | | | 5x5c3s2 | 37.0 |
| Flood | 3x3c6m5 | 10.3 | Signpost | 2x3 | 21.7 |
| | 12x12c6m5 | 63.0 | | 4x4 | 127.0 |
| Galaxies | 3x3de | 20.7 | Singles | 2x3de | 7.0 |
| | 7x7dn | 244.5 | | 5x5de | 29.5 |
| Guess | c2p3g10Bm | 21.3 | Sixteen | 2x3 | 41.0 |
| | c6p4g10Bm | 63.0 | | 3x3 | 88.5 |
| Inertia | 4x4 | 4.3 | Slant | 2x2de | 10.0 |
| | 10x8 | 33.5 | | 5x5de | 162.3 |
| Keen | 3dem | 22.3 | Solo | 2x2 | 51.3 |
| | 4de | 89.0 | | | |
| Light Up | 3x3b20s0d0 | 10.3 | Tents | 4x4de | 15.3 |
| | 7x7b20s4d0 | 115.7 | | 8x8de | 262.0 |
| Loopy | 3x3t0de | 40.7 | Towers | 4de | 27.7 |
| | 7x7t0de | 292.0 | | 4de | 104.0 |
| Magnets | 3x3deS | 13.7 | Tracks | 4x4de | 37.7 |
| | 6x5de | 103.0 | | 8x8de | 430.0 |
| Map | 3x3n5de | 6.7 | Twiddle | 2x3n2 | 22.3 |
| | 20x15n30 | 149.0 | | 3x3 rows only | 31.0 |
| Mines | 4x4n2 | 14.3 | Undead | 3x3de | 16.3 |
| | 9x9n10 | 152.0 | | 4x4de | 68.5 |
| Mosaic | 3x3 | 34.0 | Unequal | 3de | 18.7 |
| | | | | 4de | 85.0 |
| Net | 2x2 | 10.3 | Unruly | 6x6dt | 94.7 |
| | 5x5 | 125.0 | | 8x8dt | 375.5 |
| Netslide | 2x3b1 | 16.7 | Untangle | 4 | 6.0 |
| | 3x3b1 | 40.5 | | 6 | 30.5 |

## F.3  Detailed Baseline Results

We summarize all evaluated algorithms in Table 9.

*Table 9:* Summary of all evaluated RL algorithms.

| Algorithm | Policy Type | Action Masking |
|---|---|---|
| Proximal Policy Optimization (PPO) [67] | On-Policy | No |
| Recurrent PPO [68] | On-Policy | No |
| Advantage Actor Critic (A2C) [69] | On-Policy | No |
| Asynchronous Advantage Actor Critic (A3C) [69] | On-Policy | No |
| Trust Region Policy Optimization (TRPO) [70] | On-Policy | No |
| Deep Q-Network (DQN) [11] | Off-Policy | No |
| Quantile Regression DQN (QRDQN) [71] | Off-Policy | No |
| MuZero [72] | Off-Policy | Yes |
| DreamerV3 [73] | Off-Policy | No |

As we limited the agents to a single final reward upon completion, where possible, we chose puzzle parameters that allowed random policies to successfully find a solution. Note that if a random policy fails to find a solution, an RL algorithm without guidance (such as intermediate rewards) will also be affected by this. If an agent has never accumulated a reward with the initial (random) policy, it will be unable to improve its performance at all.

The chosen parameters roughly corresponded to the smallest and easiest puzzles, as more complex puzzles were found to be intractable. This fact is highlighted for example in *Solo/Sudoku*, where the reasoning needed to find a valid solution is already rather complex, even for a grid with $2\times2$ sub-blocks. A few puzzles were still intractable due to the minimum complexity permitted by Tathams's puzzle-specific problem generators, such as with *Unruly*.

For the RGB pixel observations, the window size chosen for these small problems was set at $128\times128$ pixels.

*Table 10:* Listed below are the detailed results for all evaluated algorithms. Results show the average number of steps required for all successful episodes and standard deviation with respect to the random seeds. In brackets, we show the overall percentage of successful episodes. In the summary row, the last number in brackets denotes the total number of puzzles where a solution below the upper bound of optimal steps was found. Entries without values mean that no successful policy was found among all random seeds. This Table is continued in Table 11.

| Puzzle | Supplied Parameters | Optimal | Random | PPO | TRPO | DreamerV3 | MuZero |
|---|---|---|---|---|---|---|---|
| Blackbox | w2h2m2M2 | 144 | 2206 (99.2%) | 1773 ± 472 (59.5%) | 1744 ± 454 (96.3%) | **32 ± 5** (100.0%) | **46 ± 0** (0.1%) |
| Bridges | 3x3 | 378 | 547 (100.0%) | 682 ± 197 (85.1%) | 546 ± 13 (100.0%) | **9 ± 0** (100.0%) | 397 ± 181 (86.7%) |
| Cube | c3x3 | 54 | 4181 (66.9%) | 744 ± 1610 (77.5%) | 433 ± 917 (99.8%) | 5068 ± 657 (22.5%) | - (0.0%) |
| Dominosa | 1dt | 32 | 1980 (99.2%) | 457 ± 954 (70.0%) | **12 ± 1** (100.0%) | **11 ± 1** (100.0%) | 3659 ± 0 (0.0%) |
| Fifteen | 2x2 | 256 | 54 (100.0%) | **3 ± 0** (100.0%) | 3 ± 0 (100.0%) | 4 ± 0 (100.0%) | 5 ± 1 (100.0%) |
| Filling | 2x3 | 36 | 820 (100.0%) | 290 ± 249 (97.5%) | **9 ± 2** (90.8%) | 443 ± 56 (83.4%) | 1099 ± 626 (15.0%) |
| Flip | 3x3c | 63 | 3138 (88.9%) | 3008 ± 837 (40.1%) | 2951 ± 564 (90.8%) | 1762 ± 568 (8.0%) | 1207 ± 1305 (3.1%) |
| Flood | 3x3c6m5 | 63 | 134 (97.4%) | **12 ± 0** (99.9%) | 21 ± 4 (99.6%) | **14 ± 1** (100.0%) | 994 ± 472 (14.4%) |
| Galaxies | 3x3de | 156 | 4306 (33.9%) | 3860 ± 1778 (8.3%) | 4755 ± 527 (24.8%) | 3367 ± 1585 (11.0%) | 6046 ± 2722 (8.2%) |
| Guess | c2p3g10Bm | 200 | 358 (73.4%) | | 316 ± 52 (72.0%) | 268 ± 226 (77.0%) | **24 ± 0** (0.8%) |
| Inertia | 4x4 | 51 | 13 (6.5%) | **22 ± 9** (6.3%) | 635 ± 1373 (5.7%) | 926 ± 217 (5.7%) | 104 ± 73 (3.1%) |
| Keen | 3dem | 63 | 3152 (0.5%) | 3817 ± 0 (0.2%) | 5887 ± 1526 (0.4%) | 4350 ± 1163 (1.3%) | |
| Lightup | 3x3b20s0d0 | 35 | 2237 (98.1%) | 1522 ± 1115 (82.7%) | 2127 ± 168 (95.8%) | 438 ± 247 (72.0%) | 1178 ± 1109 (2.1%) |
| Loopy | 3x3t0de | 4617 | | | | | |
| Magnets | 3x3deS | 72 | 1895 (99.1%) | 1366 ± 1090 (90.2%) | 1912 ± 60 (99.1%) | 574 ± 56 (78.5%) | 1491 ± 0 (0.7%) |
| Map | 3x3n5de | 70 | 903 (99.9%) | 1172 ± 297 (75.7%) | 950 ± 34 (99.9%) | 1680 ± 197 (64.9%) | 467 ± 328 (0.9%) |
| Mines | 4x4n2 | 144 | 87 (18.1%) | 2478 ± 2424 (9.9%) | **123 ± 66** (18.8%) | 272 ± 246 (50.1%) | **19 ± 22** (4.6%) |
| Mosaic | 3x3 | 63 | 4996 (9.8%) | 4928 ± 438 (2.5%) | 5233 ± 615 (5.0%) | 4469 ± 387 (15.9%) | 5586 ± 0 (0.2%) |
| Net | 2x2 | 28 | 1279 (100.0%) | **9 ± 0** (100.0%) | **9 ± 0** (100.0%) | **10 ± 0** (100.0%) | 339 ± 448 (8.2%) |
| Netslide | 2x3b1 | 48 | 766 (100.0%) | 1612 ± 1229 (41.6%) | 635 ± 145 (100.0%) | **12 ± 0** (100.0%) | 683 ± 810 (25.0%) |
| Netslide | 3x3b1 | 90 | 4671 (11.0%) | 4671 ± 498 (9.2%) | 4008 ± 1214 (8.9%) | 3586 ± 677 (22.4%) | 3721 ± 1461 (13.2%) |
| Palisade | 2x3n3 | 56 | 1428 (100.0%) | 939 ± 604 (87.0%) | 1377 ± 35 (99.9%) | **39 ± 56** (100.0%) | 86 ± 0 (0.0%) |
| Pattern | 3x2 | 36 | 3247 (92.9%) | 1542 ± 1262 (71.9%) | 2908 ± 355 (90.2%) | 820 ± 516 (58.0%) | 4063 ± 1696 (1.9%) |
| Pearl | 5x5de | 300 | | | | | |
| Pegs | 4x4Random | 160 | | | | | |
| Range | 3x3 | 63 | 535 (100.0%) | 780 ± 305 (65.8%) | 661 ± 198 (99.9%) | 888 ± 238 (55.6%) | 91 ± 76 (5.1%) |
| Rect | 3x2 | 72 | 723 (100.0%) | **27 ± 44** (99.8%) | **9 ± 4** (100.0%) | **8 ± 1** (100.0%) | |
| Samegame | 2x3c3s2 | 42 | 76 (100.0%) | 123 ± 197 (98.8%) | **7 ± 0** (100.0%) | **7 ± 0** (100.0%) | 1444 ± 541 (28.7%) |
| Samegame | 5x5c3s2 | 300 | 571 (32.1%) | 1003 ± 827 (30.5%) | 672 ± 160 (30.8%) | 527 ± 162 (30.2%) | **184 ± 107** (4.9%) |
| Signpost | 2x3 | 72 | 776 (96.1%) | 838 ± 53 (97.2%) | 799 ± 13 (97.0%) | 859 ± 304 (91.3%) | 4883 ± 1285 (5.9%) |
| Singles | 2x3de | 36 | 353 (100.0%) | **7 ± 3** (100.0%) | **7 ± 4** (100.0%) | **11 ± 8** (99.9%) | 733 ± 551 (28.4%) |
| Sixteen | 2x3 | 48 | 2908 (94.1%) | 2371 ± 1226 (55.7%) | 2968 ± 181 (92.8%) | **17 ± 1** (100.0%) | 3281 ± 472 (68.7%) |
| Slant | 2x2de | 20 | 447 (100.0%) | 333 ± 190 (80.4%) | 21 ± 2 (99.9%) | 596 ± 163 (100.0%) | 1005 ± 665 (7.4%) |
| Solo | 2x2 | 144 | | | | | |
| Tents | 4x4de | 56 | 4442 (44.3%) | 4781 ± 86 (10.3%) | 4828 ± 752 (31.0%) | 3137 ± 581 (12.1%) | 4556 ± 3259 (0.6%) |
| Towers | 3de | 72 | 4876 (1.0%) | | 3789 ± 1288 (0.5%) | 3746 ± 1861 (0.5%) | - |
| Tracks | 4x4de | 272 | 5213 (0.5%) | 4129 ± *nan* (0.1%) | 5499 ± 2268 (0.3%) | 4483 ± 1513 (0.3%) | - |
| Twiddle | 2x3n2 | 98 | 851 (100.0%) | **8 ± 1** (99.9%) | **11 ± 7** (100.0%) | **8 ± 0** (100.0%) | 761 ± 860 (37.6%) |
| Undead | 3x3de | 63 | 4390 (40.1%) | 4542 ± 292 (5.7%) | 4179 ± 299 (31.0%) | 4088 ± 297 (35.8%) | 3677 ± 342 (9.0%) |
| Unequal | 3de | 63 | 4540 (6.7%) | | 5105 ± 193 (3.6%) | 2468 ± 2025 (4.8%) | 4944 ± 368 (7.2%) |
| Unruly | 6x6dt | 468 | | | | | |
| Untangle | 4 | 150 | 141 (100.0%) | **13 ± 1** (100.0%) | **11 ± 0** (100.0%) | **6 ± 0** (100.0%) | 499 ± 636 (26.5%) |
| Untangle | 6 | 79 | 2165 (96.9%) | 2295 ± 66 (96.2%) | 2228 ± 126 (96.5%) | 1683 ± 74 (82.0%) | 2380 ± 0 (11.2%) |
| Summary | - | 217 | 1984 (71.2%) | 1604 ± 801 (61.6%)(8) | 1773 ± 639 (70.8%)(11) | 1334 ± 654 (62.7%)(14) | 1808 ± 983 (16.0%)(5) |

Table 11: Continuation from Table 10. Listed below are the detailed results for all evaluated algorithms. Results show the average number of steps required for all successful episodes and standard deviation with respect to the random seeds. In brackets, we show the overall percentage of successful episodes. In the summary row, the last number in brackets denotes the total number of puzzles where a solution below the upper bound of optimal steps was found. Entries without values mean that no successful policy was found among all random seeds.

| Puzzle | Supplied Parameters | Optimal | Random | A2C | RecurrentPPO | DQN | QRDQN |
|---|---|---|---|---|---|---|---|
| Blackbox | w2h2m2M2 | 144 | 2206 (99.2%) | 2524 ± 1193 (85.2%) | 2009 ± 427 (98.7%) | 2063 ± 70 (99.0%) | 2984 ± 1584 (76.8%) |
| Bridges | 3x3 | 378 | 547 (100.0%) | 540 ± 69 (100.0%) | 653 ± 165 (100.0%) | 549 ± 20 (100.0%) | 1504 ± 2037 (83.4%) |
| Cube | c3x3 | 54 | 4181 (66.9%) | 4516 ± 954 (17.5%) | 4943 ± 620 (16.2%) | 4407 ± 414 (43.4%) | 4241 ± 283 (26.4%) |
| Dominosa | 1dt | 32 | 1980 (99.2%) | 6408 ± nan (0.2%) | 3009 ± 988 (80.6%) | **15 ± 6** (100.0%) | 4457 ± 2183 (50.0%) |
| Fifteen | 2x2 | 256 | 54 (100.0%) | 4 ± 1 (100.0%) | **3 ± 0** (100.0%) | **3 ± 0** (100.0%) | **3 ± 0** (100.0%) |
| Filling | 2x3 | 36 | 820 (100.0%) | 777 ± 310 (99.3%) | 764 ± 106 (100.0%) | 761 ± 109 (99.7%) | 2828 ± 2769 (63.2%) |
| Flip | 3x3c | 63 | 3138 (88.9%) | 4345 ± 1928 (29.4%) | 3356 ± 1412 (46.9%) | 3493 ± 129 (87.1%) | 3741 ± 353 (56.8%) |
| Flood | 3x3c6m5 | 63 | 134 (97.4%) | 406 ± 623 (93.4%) | 120 ± 17 (97.7%) | 128 ± 12 (90.8%) | 1954 ± 2309 (65.2%) |
| Galaxies | 3x3de | 156 | 4306 (33.9%) | 4586 ± 980 (10.8%) | 3939 ± 1438 (0.4%) | 4657 ± 147 (26.1%) | - |
| Guess | c2p3g10Bm | 200 | 358 (73.4%) | - | 323 ± 52 (44.6%) | 550 ± 248 (71.9%) | 3260 ± 2614 (34.4%) |
| Inertia | 4x4 | 51 | 13 (6.5%) | 105 ± 197 (6.1%) | 1198 ± 1482 (5.6%) | 179 ± 156 (7.1%) | 1330 ± 296 (5.8%) |
| Keen | 3dem | 63 | 3152 (0.5%) | - | - | 6774 ± 1046 (0.4%) | - |
| Lightup | 3x3b20s0d0 | 35 | 2237 (98.1%) | 3034 ± 793 (62.7%) | 3493 ± 929 (66.5%) | 2429 ± 214 (97.5%) | 3440 ± 945 (57.8%) |
| Loopy | 3x3t0de | 4617 | - | - | - | - | - |
| Magnets | 3x3deS | 72 | 1895 (99.1%) | 3057 ± 1114 (47.9%) | 1874 ± 222 (99.2%) | 2112 ± 331 (98.1%) | 5182 ± 3878 (33.8%) |
| Map | 3x3n5de | 70 | 903 (99.9%) | 2552 ± 1223 (52.5%) | 2608 ± 1808 (59.4%) | 949 ± 30 (99.9%) | 1753 ± 769 (78.1%) |
| Mines | 4x4n2 | 144 | 87 (18.1%) | **120 ± 41** (14.7%) | 1189 ± 1341 (12.1%) | 207 ± 146 (17.6%) | 1576 ± 1051 (13.2%) |
| Mosaic | 3x3 | 63 | 4996 (9.8%) | 4937 ± 424 (8.4%) | 4907 ± 219 (8.3%) | 5279 ± 564 (7.0%) | 9490 ± 155 (0.0%) |
| Net | 2x2 | 28 | 1279 (100.0%) | 149 ± 288 (100.0%) | 1232 ± 92 (100.0%) | **9 ± 0** (100.0%) | 1793 ± 1663 (81.3%) |
| Netslide | 2x3b1 | 48 | 766 (100.0%) | 976 ± 584 (100.0%) | 2079 ± 1989 (64.7%) | 7779 ± 37 (100.0%) | 1023 ± 206 (80.9%) |
| Netslide | 3x3b1 | 90 | 4671 (11.0%) | 4324 ± 657 (8.1%) | 2737 ± 1457 (1.7%) | 4099 ± 846 (5.1%) | 2025 ± 1475 (0.4%) |
| Palisade | 2x3n3 | 56 | 1428 (100.0%) | 1666 ± 198 (99.4%) | 1981 ± 1053 (92.5%) | 1445 ± 96 (99.9%) | 1519 ± 142 (99.8%) |
| Pattern | 3x2 | 36 | 3247 (92.9%) | 3445 ± 635 (82.9%) | 3733 ± 513 (79.7%) | 2809 ± 733 (89.7%) | 3406 ± 384 (51.1%) |
| Pearl | 5x5de | 300 | - | - | - | - | - |
| Pegs | 4x4Random | 160 | - | - | - | - | - |
| Range | 3x3 | 63 | 535 (100.0%) | 1438 ± 782 (81.4%) | 730 ± 172 (99.9%) | 594 ± 28 (100.0%) | 1560 ± 1553 (81.8%) |
| Rect | 3x2 | 72 | 723 (100.0%) | 3470 ± 2521 (17.6%) | 916 ± 420 (99.6%) | 511 ± 193 (97.4%) | **14 ± 9** (100.0%) |
| Samegame | 2x3c3s2 | 42 | 76 (100.0%) | 8 ± 1 (100.0%) | 1777 ± 1643 (43.5%) | **8 ± 0** (100.0%) | 5577 ± 1211 (12.8%) |
| Samegame | 5x5c3s2 | 300 | 571 (32.1%) | 609 ± 155 (29.9%) | 1321 ± 1170 (30.3%) | 850 ± 546 (29.2%) | 2298 ± 2845 (78.0%) |
| Signpost | 2x3 | 72 | 776 (96.1%) | 2259 ± 1394 (85.9%) | 1000 ± 266 (77.9%) | 793 ± 17 (97.0%) | 392 ± 29 (100.0%) |
| Singles | 2x3de | 36 | 353 (100.0%) | 372 ± 47 (100.0%) | 331 ± 66 (100.0%) | 361 ± 47 (99.1%) | 4550 ± 848 (21.9%) |
| Sixteen | 2x3 | 48 | 2908 (94.1%) | 3903 ± 479 (71.7%) | 3409 ± 574 (67.6%) | 2970 ± 107 (93.2%) | 1398 ± 2097 (87.1%) |
| Slant | 2x2de | 20 | 447 (100.0%) | 984 ± 470 (99.8%) | 465 ± 34 (100.0%) | 496 ± 97 (100.0%) | 5295 ± 688 (7.8%) |
| Solo | 2x2 | 144 | - | - | - | - | - |
| Tents | 4x4de | 56 | 4442 (44.3%) | 6157 ± 1961 (2.1%) | 4980 ± 397 (12.8%) | 4515 ± 59 (38.1%) | 3170 ± 1479 (33.4%) |
| Towers | 3de | 72 | 4876 (1.0%) | 9850 ± nan (0.0%) | 8549 ± nan (0.0%) | 5836 ± 776 (0.5%) | - |
| Tracks | 4x4de | 272 | 5213 (0.5%) | 4501 ± nan (0.0%) | - | 5809 ± 661 (0.3%) | - |
| Twiddle | 2x3n2 | 98 | 851 (100.0%) | 1248 ± 430 (99.6%) | 827 ± 71 (100.0%) | **83 ± 149** (100.0%) | 871 ± 837 (99.0%) |
| Undead | 3x3de | 63 | 4390 (40.1%) | 5818 ± 154 (0.9%) | 5060 ± 2381 (0.5%) | - | - |
| Unequal | 3de | 63 | 4540 (6.7%) | 5067 ± 1600 (1.0%) | 5929 ± 1741 (1.1%) | 5057 ± 582 (5.6%) | - |
| Unruly | 6x6d4t | 468 | - | - | - | - | - |
| Untangle | 4 | 150 | 141 (100.0%) | 1270 ± 1745 (90.4%) | **135 ± 18** (100.0%) | 170 ± 29 (100.0%) | - |
| Untangle | 6 | 79 | 2165 (96.9%) | 3324 ± 1165 (72.5%) | 2739 ± 588 (91.7%) | 2219 ± 84 (95.9%) | - |
| Summary | - | 217 | 1984 (71.2%) | 2743 ± 954 (54.8%)(3) | 2342 ± 989 (61.1%)(2) | 1999 ± 365 (70.2%)(5) | 2754 ± 1579 (56.0%)(2) |

*Table 12:* We list the detailed results for all the experiments of action masking and input representation. Results show the average number of steps required for all successful episodes and standard deviation with respect to the random seeds. In brackets, we show the overall percentage of successful episodes. In the summary row, the last number in brackets denotes the total number of puzzles where a solution below the upper bound of optimal steps was found. Entries without values mean that no successful policy was found among all random seeds.

| Puzzle | Supplied Parameters | Optimal | Random | PPO (Internal State) | PPO (RGB Pixels) | MaskablePPO (Internal State) | MaskablePPO (RGB Pixels) |
|---|---|---|---|---|---|---|---|
| Blackbox | w2h2m2M2 | 144 | 2206 (99.2%) | 1773 ± 472 (59.5%) | 1509 ± 792 (97.9%) | **9 ± 0** (99.7%) | **30 ± 1** (99.2%) |
| Bridges | 3x3 | 378 | 547 (100.0%) | 682 ± 197 (85.1%) | **89 ± 176** (99.1%) | **25 ± 0** (99.4%) | **9 ± 0** (99.6%) |
| Cube | c3x3 | 54 | 4181 (66.9%) | 744 ± 1610 (77.5%) | 3977 ± 442 (67.7%) | **16 ± 1** (81.2%) | 410 ± 157 (75.1%) |
| Dominosa | 1dt | 32 | 1980 (99.2%) | 457 ± 954 (70.0%) | 539 ± 581 (100.0%) | **12 ± 0** (100.0%) | **19 ± 2** (100.0%) |
| Fifteen | 2x2 | 256 | 54 (100.0%) | **3 ± 0** (100.0%) | **37 ± 26** (100.0%) | **4 ± 0** (100.0%) | **3 ± 0** (100.0%) |
| Filling | 2x3 | 36 | 820 (88.9%) | 290 ± 249 (97.5%) | 373 ± 175 (99.9%) | **7 ± 0** (100.0%) | **34 ± 3** (99.9%) |
| Flip | 3x3c | 63 | 3138 (97.4%) | 3008 ± 837 (40.1%) | 3616 ± 395 (78.3%) | 2174 ± 1423 (70.3%) | 319 ± 128 (81.3%) |
| Flood | 3x3c6m5 | 63 | 134 (97.4%) | **12 ± 0** (99.9%) | **28 ± 12** (99.7%) | **12 ± 0** (99.9%) | **14 ± 0** (99.9%) |
| Galaxies | 3x3de | 156 | 4306 (33.9%) | 3860 ± 1778 (8.3%) | 4439 ± 224 (29.1%) | 3640 ± 928 (40.2%) | 3372 ± 430 (40.5%) |
| Guess | c2p3g10Bm | 200 | 358 (73.4%) | | 344 ± 35 (72.0%) | **145 ± 19** (75.4%) | |
| Inertia | 4x4 | 51 | 13 (6.5%) | **22 ± 9** (6.3%) | 237 ± 10 (99.7%) | 41 ± 19 (79.0%) | 169 ± 233 (69.8%) |
| Keen | 3dem | 63 | 3152 (0.5%) | 3817 ± 0 (0.2%) | | | |
| Lightup | 3x3b20s0d0 | 35 | 2237 (98.1%) | 1522 ± 1115 (82.7%) | 2401 ± 148 (97.5%) | **25 ± 8** (99.1%) | 1608 ± 1144 (90.1%) |
| Loopy | 3x3t0de | 4617 | | | | | |
| Magnets | 3x3deS | 72 | 1895 (99.1%) | 1366 ± 1090 (90.2%) | 1794 ± 109 (98.7%) | 222 ± 33 (98.8%) | 425 ± 68 (99.2%) |
| Map | 3x3n5de | 70 | 903 (99.9%) | 1172 ± 297 (75.7%) | 958 ± 33 (99.9%) | 321 ± 33 (99.9%) | 467 ± 69 (99.1%) |
| Mines | 4x4n2 | 144 | 87 (18.1%) | 2478 ± 2424 (9.9%) | 2406 ± 296 (44.7%) | 412 ± 268 (43.3%) | 653 ± 396 (43.1%) |
| Mosaic | 3x3 | 63 | 4996 (9.8%) | 4928 ± 438 (2.5%) | 5673 ± 1547 (6.7%) | 3381 ± 906 (29.4%) | 3158 ± 247 (28.5%) |
| Net | 2x2 | 28 | 1279 (100.0%) | **9 ± 0** (100.0%) | 180 ± 44 (100.0%) | 9 ± 0 (100.0%) | 96 ± 7 (100.0%) |
| Netslide | 2x3b1 | 48 | 766 (100.0%) | 1612 ± 1229 (41.6%) | **35 ± 18** (100.0%) | 13 ± 0 (100.0%) | |
| Netslide | 3x3b1 | 90 | 4671 (11.0%) | 4671 ± 498 (9.2%) | | | |
| Palisade | 2x3n3 | 56 | 1428 (100.0%) | 939 ± 604 (87.0%) | 1412 ± 23 (99.9%) | 90 ± 55 (99.9%) | 347 ± 26 (99.8%) |
| Pattern | 3x2 | 36 | 3247 (92.9%) | 1542 ± 1262 (71.9%) | 2983 ± 173 (92.5%) | **14 ± 0** (96.9%) | 1201 ± 1021 (88.7%) |
| Pearl | 5x5de | 300 | | | | | |
| Pegs | 4x4Random | 160 | | | | 1730 ± 579 (34.9%) | 1482 ± 687 (37.3%) |
| Range | 3x3 | 63 | 535 (100.0%) | 780 ± 305 (65.8%) | 613 ± 25 (100.0%) | **50 ± 69** (100.0%) | 209 ± 26 (100.0%) |
| Rect | 3x2 | 72 | 723 (100.0%) | **27 ± 44** (99.8%) | 300 ± 387 (100.0%) | **8 ± 0** (100.0%) | **38 ± 9** (100.0%) |
| Samegame | 2x3c3s2 | 42 | 76 (100.0%) | 123 ± 197 (98.8%) | **11 ± 8** (100.0%) | **8 ± 0** (100.0%) | **9 ± 0** (100.0%) |
| Samegame | 5x5c3s2 | 300 | 571 (32.1%) | 1003 ± 827 (30.5%) | | | |
| Signpost | 2x3 | 72 | 776 (96.1%) | 838 ± 53 (97.2%) | 779 ± 50 (97.0%) | 567 ± 149 (97.7%) | 454 ± 50 (97.5%) |
| Singles | 2x3de | 36 | 353 (100.0%) | **7 ± 3** (100.0%) | 306 ± 57 (100.0%) | **5 ± 1** (100.0%) | 218 ± 17 (100.0%) |
| Sixteen | 2x3 | 48 | 2908 (94.1%) | 2371 ± 1226 (55.7%) | 3211 ± 450 (89.6%) | **19 ± 2** (94.3%) | 3650 ± 190 (68.5%) |
| Slant | 2x2de | 20 | 447 (100.0%) | 333 ± 190 (80.4%) | 325 ± 119 (100.0%) | **12 ± 0** (100.0%) | 89 ± 21 (100.0%) |
| Solo | 2x2 | 144 | | | | | |
| Tents | 4x4de | 56 | 4442 (44.3%) | 4781 ± 86 (10.3%) | 4493 ± 155 (37.5%) | 3485 ± 63 (39.9%) | 3485 ± 456 (45.0%) |
| Towers | 3de | 72 | 4876 (1.0%) | | | | |
| Tracks | 4x4de | 272 | 5213 (0.5%) | 4129 ± nan (0.1%) | 4217 ± nan (1.6%) | 5461 ± 976 (0.3%) | 5019 ± 2297 (0.4%) |
| Twiddle | 2x3n2 | 98 | 851 (100.0%) | **8 ± 1** (99.9%) | 348 ± 466 (100.0%) | **7 ± 0** (100.0%) | 12 ± 1 (100.0%) |
| Undead | 3x3de | 63 | 4390 (40.1%) | 4542 ± 292 (5.7%) | 4129 ± 139 (40.0%) | 3415 ± 379 (42.8%) | 3482 ± 406 (46.1%) |
| Unequal | 3de | 63 | 4540 (6.7%) | | | 2322 ± 988 (38.7%) | 3021 ± 1368 (26.5%) |
| Unruly | 6x6dt | 468 | | | | | |
| Untangle | 4 | 150 | 141 (100.0%) | **13 ± 1** (100.0%) | **35 ± 58** (100.0%) | **12 ± 0** (100.0%) | **7 ± 0** (100.0%) |
| Untangle | 6 | 79 | 2165 (96.9%) | 2295 ± 66 (96.2%) | | | |
| Summary | - | 217 | 1984 (71.2%) | 1604 ± 801 (61.6%)(8) | 1619 ± 380 (82.8%)(6) | 814 ± 428 (81.2%)(21) | 1047 ± 583 (79.2%)(10) |

### F.4 Episode Length and Early Termination Parameters

In Table 13, the puzzles and parameters used for training the agents for the ablation in Section 3.4 are shown in combination with the results. Due to limited computational budget, we included only a subset of all puzzles at the easy human difficulty preset for DreamerV3. Namely, we have selected all puzzles where a random policy was able to complete at least a single episode successfully within 10,000 steps in 1000 evaluations. It contains a subset of the more challenging puzzles, as can be seen by the performance of many algorithms in Table 10. For some puzzles, e.g. Netslide, Samegame, Sixteen and Untangle, terminating episodes early brings a benefit in final evaluation performance when using a large maximal episode length during training. For the smaller maximal episode length, the difference is not always as pronounced.

*Table 13:* Listed below are the puzzles and their corresponding supplied parameters. For each setting, we report average success episode length with standard deviation with respect to the random seed, all averaged over all selected puzzles. In brackets, the percentage of successful episodes is reported. # Steps stands for the maximal episode length, ET stands for early termination after a given number of state repeats.

| Puzzle | Supplied Parameters | # Steps | ET | DreamerV3 |
|---|---|---|---|---|
| Bridges | 7x7i30e10m2d0 | $1e4$ | 10 | $4183.0 \pm 2140.5$ (0.2%) |
| | | | - | - |
| | | $1e5$ | 10 | $4017.9 \pm 1390.1$ (0.3%) |
| | | | - | $4396.2 \pm 2517.2$ (0.3%) |
| Cube | c4x4 | $1e4$ | 10 | $21.9 \pm 1.4$ (100.0%) |
| | | | - | $21.4 \pm 0.9$ (100.0%) |
| | | $1e5$ | 10 | $22.6 \pm 2.0$ (100.0%) |
| | | | - | $21.3 \pm 1.2$ (100.0%) |
| Flood | 12x12c6m5 | $1e4$ | 10 | - |
| | | | - | - |
| | | $1e5$ | 10 | - |
| | | | - | - |
| Guess | c6p4g10Bm | $1e4$ | 10 | - |
| | | | - | $1060.4 \pm 851.3$ (0.6%) |
| | | $1e5$ | 10 | $2405.5 \pm 2476.4$ (0.5%) |
| | | | - | $3165.2 \pm 1386.8$ (0.6%) |
| Netslide | 3x3b1 | $1e4$ | 10 | $3820.3 \pm 681.0$ (18.4%) |
| | | | - | $3181.3 \pm 485.5$ (21.1%) |
| | | $1e5$ | 10 | $3624.9 \pm 746.5$ (23.0%) |
| | | | - | $4050.6 \pm 505.5$ (10.6%) |
| Samegame | 5x5c3s2 | $1e4$ | 10 | $53.8 \pm 7.5$ (38.3%) |
| | | | - | $717.4 \pm 309.0$ (29.1%) |
| | | $1e5$ | 10 | $47.3 \pm 6.6$ (36.7%) |
| | | | - | $1542.9 \pm 824.0$ (26.4%) |
| Signpost | 4x4c | $1e4$ | 10 | $6848.9 \pm 677.7$ (1.1%) |
| | | | - | $6861.8 \pm 301.8$ (1.5%) |
| | | $1e5$ | 10 | $6983.7 \pm 392.4$ (1.6%) |
| | | | - | - |
| Sixteen | 3x3 | $1e4$ | 10 | $4770.5 \pm 890.5$ (2.9%) |
| | | | - | $4480.5 \pm 2259.3$ (25.5%) |
| | | $1e5$ | 10 | $3193.3 \pm 2262.0$ (57.0%) |
| | | | - | $3517.1 \pm 1846.7$ (23.5%) |
| Undead | 4x4de | $1e4$ | 10 | $5378.0 \pm 1552.7$ (0.5%) |
| | | | - | $5324.4 \pm 557.9$ (0.6%) |
| | | $1e5$ | 10 | $5666.2 \pm 553.3$ (0.5%) |
| | | | - | $5771.3 \pm 2323.6$ (0.4%) |
| Untangle | 6 | $1e4$ | 10 | $474.7 \pm 117.6$ (99.1%) |
| | | | - | $1491.9 \pm 193.8$ (89.3%) |
| | | $1e5$ | 10 | $597.0 \pm 305.5$ (96.3%) |
| | | | - | $1338.4 \pm 283.6$ (88.7%) |

### F.5 LLM Evaluation

We have extended the PUZZLES library with an interface for large language models (LLMs) and vision language models (VLMs). This allows for easy evaluation of these models on truly out-of-distribution data. Thanks to the scalability of our benchmark, puzzle sizes can be continuously increased to match the current capabilities of the models. Currently, we focus on zero-shot evaluation of LLMs/VLMs, where the models are not provided with any specific training on the puzzle they need to solve. This is similar to a human solving a puzzle for the first time. For future evaluations, however, this could be improved by either training the model on a smaller version of the puzzle, or giving it some examples and allowing it to explicitly come up with a strategy. The LLM/VLM had to play the puzzle using the same cursor interface as the RL agents and the human expert, requiring it to plan ahead and execute single steps. This also means that the model was not able to solve the puzzle directly by outputting a solution in text format.

To reduce the computational cost of LLM evaluation, we implemented early termination after state repetition. Specifically, if an LLM enters the exact same state five times, indicating no progress in solving the puzzle, the evaluation terminates. Additionally, for each puzzle, we set the maximum number of steps to the upper bound of an optimal policy.

During evaluation, the LLM is not provided with the complete history of all past actions and states. Instead, it receives only the explanation of the game, the current discrete state, an image of the current puzzle state, and the most recent past action. Action masking is employed to limit the LLM's choices to reasonable actions.

The game explanations and instructions for playing the game using the keyboard are sourced from `https://www.chiark.greenend.org.uk/~sgtatham/puzzles/doc/`. Further details about the LLM evaluation framework are available on the official GitHub repository at `https://github.com/ETH-DISCO/rlp/tree/main/llm`. The code is designed to easily support the addition and evaluation of new LLMs. For more information, please refer to `https://github.com/ETH-DISCO/rlp?tab=readme-ov-file#run-an-llm-that-is-available-via-api`.

We conducted experiments with Gemini 1.5 Flash and GPT-4o mini, allowing each LLM five attempts to solve each puzzle. Our results show that GPT-4o mini solved slightly more puzzles than Gemini 1.5 Flash. Interestingly, the evaluation with GPT-4o-mini took approximately 5 hours, while Gemini 1.5 Flash required only 1 hour.

Table 14: Success rates of LLMs on the easiest setting for all puzzles.

| Puzzle | Supplied Parameters | Gemini 1.5 Flash Success Rate (%) | GPT-4o mini Success Rate (%) |
|---|---|---|---|
| Blackbox | w2h2m2M2 | 0 | 0 |
| Bridges | 3x3 | 0 | 0 |
| Cube | c3x3 | 0 | 0 |
| Dominosa | 1dt | 0 | 0 |
| Fifteen | 2x2 | 20 | 100 |
| Fifteen | 3x3n5de | 0 | 0 |
| Filling | 2x3 | 0 | 0 |
| Flip | 3x3c | 0 | 0 |
| Flood | 3x3c6m5 | 20 | 20 |
| Galaxies | 3x3de | 0 | 0 |
| Guess | c2p3g10Bm | 0 | 0 |
| Inertia | 4x4 | 0 | 0 |
| Keen | 3dem | 0 | 0 |
| Lightup | 3x3b20s0d0 | 0 | 0 |
| Loopy | 3x3t0de | 0 | 0 |
| Magnets | 3x3deS | 0 | 20 |
| Map | 3x3n5de | 0 | 0 |
| Mines | 4x4n2 | 0 | 0 |
| Mosaic | 3x3 | 0 | 0 |
| Net | 2x2 | 0 | 0 |
| Netslide | 2x3b1 | 0 | 0 |
| Palisade | 2x3n3 | 0 | 0 |
| Pattern | 3x2 | 0 | 0 |
| Pearl | 5x5de | 0 | 0 |
| Pegs | 4x4random | 0 | 0 |
| Range | 3x3 | 0 | 0 |
| Rect | 3x2 | 0 | 0 |
| Samegame | 2x3c3s2 | 0 | 40 |
| Signpost | 2x3 | 0 | 0 |
| Singles | 2x3de | 0 | 0 |
| Sixteen | 2x3 | 0 | 0 |
| Slant | 2x2de | 0 | 0 |
| Solo | 2x2 | 0 | 0 |
| Tents | 4x4de | 0 | 0 |
| Towers | 3de | 0 | 0 |
| Tracks | 4x4de | 0 | 0 |
| Twiddle | 2x3n2 | 0 | 20 |
| Undead | 3x3de | 0 | 0 |
| Unequal | 3de | 0 | 0 |
| Unruly | 6x6dt | 0 | 0 |
| Untangle | 4 | 0 | 0 |

