# OpenReview forum: "PUZZLES: A Benchmark for Neural Algorithmic Reasoning"
_NeurIPS.cc/2024/Datasets_and_Benchmarks_Track — NeurIPS 2024 Track Datasets and Benchmarks Poster_

### Official Review · Reviewer_EJo1 · 2024-07-24
**Puzzle benchmark for reinforcement learning**

**Rating:** 7
**Confidence:** 4
**Clarity:** Paper is well written and throughout …

**Review:**

The paper presents a focused, fast and light set of benchmarks for testing RL agents and their capability of learning to do logical reasoning/planning in the puzzles, which can be tested by evaluating the generalization to different scales of the same problem. This benchmark will help to study a specific aspect of RL algorithms.

However, given the nature of the benchmark (focus on "logical thinking" / "planning" / "reasoning"), RL feels bit misleading domain to use these puzzles for. We can come up with RL algorithms that solve these puzzles, but unless they are also scalable and trainable in more complex domains (e.g., Minecraft), I believe the impact of those agents (and thus this benchmark) is limited, as this encourages coming up with solutions that may only work in this domain.

To make this benchmark more significant, it should provide interfaces for large language-models to also evaluate their capabilities to solve the puzzles (if LLMs truly capture "planning" and "logical thinking", we may start seeing monumental shifts in society, unlike with RL algorithms).

In addition, given the simplicity of the puzzles, these environments run bit slower than I would expect (few hours to train an agent). For a very significant help in RL community, experiments that can be run in tens of minutes on single GPU / CPUs would be very useful. For example, people can run MuJoCo-like experiments with [Brax](https://github.com/google/brax) or [Jax stable-baselines](https://github.com/araffin/sbx) in less than a hour on the right setup. However, the benefit of PUZZLEs is that by not using Jax or other vectorized language, it will run fine as a single environment (where-as Jax versions might become even slower).

* Quality: Paper and code are of high quality, and come with extensive details on the environment definition and how to run the experiments in the paper.

* Clarity: Paper is well written. No comments.

* Originality: Medium/High. While puzzles have been around in many forms (e.g., original Gym had many smaller tasks), they did not use this set of puzzles made for humans (Simon Tatham's Portable Puzzle Collection).

* Significance: Medium. The logical reasoning and ability to test in larger scales or higher difficulty make this benchmark good for testing if the RL algorithm has truly figured out the "rules" of the puzzle. However, I feel the signifcance is limited if focused on RL solely. If the environment offered also interfaces for large language models to interface with the puzzles and try to solve them, then they could be used to test the logical reasoning of LLMs as well.

**Strengths:**

- Fast implementations of set of puzzles to test logical reasoning of RL agents.
- Benchmark allow parametrizing the environments with different difficulties and/or scale, so that researchers can increase the difficulty of the task or test generalization (e.g., did the agent really learn how to solve the puzzle)
- Benchmarks with standard RL agents + set of ablations with different difficulties, observation and action spaces and in a generalization domain. The results show that the agents fall short.

**Additional Feedback:**

-

**Correctness:**

Experiments are ran with five random seeds for statistical significance (although see "opportunities for improvement" on the statistical significance).

**Documentation:**

Authors provide the code and steps to reproduce the experiments in the github repository, which also works as an example of how to run experiments with the library.

**Ethics:**

No ethical concerns.

**Limitations:**

- As stated by the authors, the discrete nature of these benchmarks limit the applicable algorithms to ones with discrete outputs. Granted, this is more of an issue of broader RL community, with different groups focusing on different domains (robotics vs. games, for example).

- Also stated by the authors, the results do not contain hyperparameter sweeps or any optimizations for solving these tasks. However, for a benchmark paper, this is ok.

- This benchmark would also be valuable for the LLM community where the models are at the stage that they could solve puzzles like this and learn/capture the real logic behind them. Even in the best case, RL agents of this format, will not have such big impact on the world for years to come, as they are still very single-task oriented and lacking in complexity. Meanwhile, LLM space is currently trying working on improving the "reasoning", "logical thinking" and "planning" of the agents, and the more benchmarks they have, the better signal there is that we are going the right way. I believe this benchmark could have more significant impact with an LLM interface.

- The "Noncommercial" (NC) part of the provided license restricts the use of the library by companies. While it does not explicitely prohibit the use in a big company, company policies may prevent use of any NC licensed tools as the NC-nature might "contaminate" any results or codebases. At least two large AI companies would prohibit use of this library because of the license, or at least cause additional steps to work with the library. While I understand the desire to license code in way to prevent others from "abusing" it for their own good, I encourage authors to rethink the license given this information.

**Opportunities For Improvement:**

* The performance of agents is shown as "average episode length", with lower being better (agent solves the task faster). This skews the results bit, as it is heavily dependent on how what is the upper bound for episode length. I would also recommend reporting the success rates (i.e., given maximum number of steps, how often the agent solved the solution within the time), to give a better view.

* Results in Figure 3 have very high standard deviations, and with five random seeds, I do not think the means are not significantly different (i.e., the results of the agents are not significantly different). I would suggest authors do run some statistical tests (e.g., usual t-test), and revise the conclusions.

* Additionally, given that the results are aggregated over multiple environments, I suggest authors to check out "rliable" library for more informative ways of aggregating results over multiple environments rather than averaging: https://github.com/google-research/rliable

### Questions

1) How fast are the environments to run without training (i.e., the raw throughput), and especially with GPUs vs. CPUs? And how many environments people can run in parallel? I reckon these environments are very fast, and with some optimization (e.g., Jax-ifying the code or somehow integrating with the training agent), you could train these agents in matter of minutes instead of hours. This would unlock new ways of doing research with these environments (e.g., huge sweeps with relatively small hardware).

2) Table 3: the "PPO (Transformer)\cross" is said to be the checkpoint that has the highest performance in the generalization environment, but in the table we have higher steps to complete episode but higher completion rates. Am I reading this correctly? I would have expected both episode length to go down and the success rate go up with cherry-picked model.

3) Would it be possible to conduct experiments with an LLM, such as recently released Llama 3.1 (https://llama.meta.com/) or GPT-4, by creating an interface that allows playing the games through text? While this is a significant detour from the RL focus of the library, these puzzles could provide signal how well the LLMs are able to "reason" and plan ahead to solve the puzzles. I feel the library would have more significant impact if it had pre-made interfaces for LLMs (in addition to RL agents), and had initial benchmarks with a simple/trivial setup.

**Relation To Prior Work:**

Authors discuss the previous works on RL benchmarks, and specifically RL benchmarks on logical thinking and reasoning. However, there is not clear connection how this benchmark adds on top of these benchmarks, other than by being an unique set.

**Summary And Contributions:**

Authors present PUZZLES benchmark, based on a set of existing puzzle games (designed for people), to challenge the logical reasoning of reinforcement learning (RL) agents. The benchmark allows adjusting the scale and difficulty of the puzzles. Authors then run baseline benchmarks with RL agents, and study how changing the task difficulty, the observation space, use of action masking and use of early termination impacts the results. Finally, authors also study generalization of the agents to different scales.

---

> ### Author Rebuttal · Authors · 2024-08-16
>
> We thank the reviewer for their effort on reviewing our work. We address their concerns in the following.
>
> > The performance of agents is shown as "average episode length", with lower being better (agent solves the task faster). This skews the results bit, as it is heavily dependent on how what is the upper bound for episode length. I would also recommend reporting the success rates (i.e., given maximum number of steps, how often the agent solved the solution within the time), to give a better view.
>
> We will update the manuscript with another plot containing the success rate to Figure 3. Currently, success rate and number of puzzles solved within the upper bound on optimal number of steps is reported in the appendix in Tables 9 and 10.
>
> > Results in Figure 3 have very high standard deviations, and with five random seeds, I do not think the means are not significantly different (i.e., the results of the agents are not significantly different). I would suggest authors do run some statistical tests (e.g., usual t-test), and revise the conclusions.
>
> The high standard deviation indicates that this particular algorithm struggled to consistently learn solutions across different random seeds. While the results may not be statistically significant when aggregated across all puzzles, a closer look at individual puzzles reveals that some algorithms perform significantly better than others. We are currently revising the manuscript to provide more precise and accurate details in this regard.
>
>
> > Additionally, given that the results are aggregated over multiple environments, I suggest authors to check out "rliable" library for more informative ways of aggregating results over multiple environments rather than averaging: https://github.com/google-research/rliable
>
> We thank the reviewer for this valuable suggestion. We are in the process of integrating rliable and will update the figures in our manuscript accordingly.
>
> > How fast are the environments to run without training (i.e., the raw throughput), and especially with GPUs vs. CPUs? And how many environments people can run in parallel? I reckon these environments are very fast, and with some optimization (e.g., Jax-ifying the code or somehow integrating with the training agent), you could train these agents in matter of minutes instead of hours. This would unlock new ways of doing research with these environments (e.g., huge sweeps with relatively small hardware).
>
> The environments are designed to run on CPU only, so unfortunately, using a GPU currently does not provide any performance advantage. The raw throughput varies depending on the puzzle but typically ranges between 5,000 and 10,000 steps per second. Due to the use of the Pygame interface, parallelization via vectorized environments is not straightforward. While converting the codebase to JAX would be ideal, doing so with Simon Tatham’s codebase would require a substantial additional time investment. By open-sourcing our codebase, we hope the community can contribute to this effort.
>
> > Table 3: the "PPO (Transformer)\cross" is said to be the checkpoint that has the highest performance in the generalization environment, but in the table we have higher steps to complete episode but higher completion rates. Am I reading this correctly? I would have expected both episode length to go down and the success rate go up with cherry-picked model.
>
> We observe that strategies which need fewer steps in the training environment don’t generalize well to the generalization environment. However, these findings should be interpreted with caution since the transformer models generally do not learn strong policies.
>
> > Would it be possible to conduct experiments with an LLM, such as recently released Llama 3.1 (https://llama.meta.com/) or GPT-4, by creating an interface that allows playing the games through text? While this is a significant detour from the RL focus of the library, these puzzles could provide signal how well the LLMs are able to "reason" and plan ahead to solve the puzzles. I feel the library would have more significant impact if it had pre-made interfaces for LLMs (in addition to RL agents), and had initial benchmarks with a simple/trivial setup.
>
> We agree that our logic puzzle benchmark could also provide interesting and valuable insights for LLM research. We will publish an interface that allows for direct integration and evaluation of LLMs.
>
>
> > The "Noncommercial" (NC) part of the provided license restricts the use of the library by companies. While it does not explicitely prohibit the use in a big company, company policies may prevent use of any NC licensed tools as the NC-nature might "contaminate" any results or codebases. At least two large AI companies would prohibit use of this library because of the license, or at least cause additional steps to work with the library. While I understand the desire to license code in way to prevent others from "abusing" it for their own good, I encourage authors to rethink the license given this information.
>
> Thank you for this remark. We initially believed [Simon Tatham’s license](https://www.chiark.greenend.org.uk/~sgtatham/puzzles/doc/licence.html) was restrictive and would constrain us to an NC license but it turns out it is not. We have updated our license to an MIT license.

---

> > ### Comment · Reviewer_EJo1 · 2024-08-23
> >
> > Thank you for your replies! Overall I am satisfied with the answers, and have increased rating from 6 to 7, assuming the manuscript will be updated as described. This environment provides good way to analyze the generalization of RL (and LLM) agents with the scale and difficulty settings, and I see it being useful for years to come.
> >
> > > Jax / GPU environments
> >
> > 5k/10k steps per second already sounds high enough, especially if things can be parallelized. I also just realized that, because of the nature of these puzzles, they may not scale well on GPUs because of amount of branching (lots of conditional branches will lead to poor GPU utilization).
> >
> > > License
> >
> > Thank you for updating this! It can seem like an annoying technical nitpick, but licenses exist for a reason.

---

> > > ### Author Response · Authors · 2024-08-31
> > >
> > > Thank you for your reply and for increasing your score!
> > >
> > > > We agree that our logic puzzle benchmark could also provide interesting and valuable insights for LLM research. We will publish an interface that allows for direct integration and evaluation of LLMs.
> > >
> > > We have now updated our [Github](https://github.com/ETH-DISCO/rlp) with an interface and sample scripts for integrating and evaluating LLMs, see section *Run an LLM that is available via API* in the README. Thanks to the standard gymnasium interface of the PUZZLES library, works such as [LlamaGym](https://github.com/KhoomeiK/LlamaGym) can easily be used to fine-tune LLMs on our puzzles.
> > >
> > > Furthermore, we performed baseline evaluations with 5 random seeds on the easiest puzzle settings for both GPT-4o Mini and Gemini 1.5 Flash. We provide an overview of the results in the following table. We will include these results in a more detailed form to the appendix of our manuscript. We can see that GPT-4o Mini was able to solve slightly more puzzles than Gemini 1.5 Flash. Interestingly, running the evaluation with GPT4o-mini took approximately 5 hours while Gemini 1.5 Flash took only 1 hour.
> > >
> > > | Puzzle   | Arg        | Gemini Success Rate (%)   | GPT Success Rate (%)   |
> > > |:---------|:-----------|:--------------------------|:-----------------------|
> > > | blackbox | w2h2m2M2   | 0                         | 0                      |
> > > | bridges  | 3x3        | 0                         | 0                      |
> > > | cube     | c3x3       | 0                         | 0                      |
> > > | dominosa | 1dt        | 0                         | 0                      |
> > > | fifteen  | 2x2        | 20                        | 100                    |
> > > | filling  | 2x3        | 0                         | 0                      |
> > > | flip     | 3x3c       | 0                         | 0                      |
> > > | flood    | 3x3c6m5    | 20                        | 20                     |
> > > | galaxies | 3x3de      | 0                         | 0                      |
> > > | guess    | c2p3g10Bm  | 0                         | 0                      |
> > > | inertia  | 4x4        | 0                         | 0                      |
> > > | keen     | 3dem       | 0                         | 0                      |
> > > | lightup  | 3x3b20s0d0 | 0                         | 0                      |
> > > | loopy    | 3x3t0de    | 0                         | 0                      |
> > > | magnets  | 3x3deS     | 0                         | 20                     |
> > > | map      | 3x3n5de    | 0                         | 0                      |
> > > | mines    | 4x4n2      | 0                         | 0                      |
> > > | mosaic   | 3x3        | 0                         | 0                      |
> > > | net      | 2x2        | 0                         | 0                      |
> > > | netslide | 2x3b1      | 0                         | 0                      |
> > > | palisade | 2x3n3      | 0                         | 0                      |
> > > | pattern  | 3x2        | 0                         | 0                      |
> > > | pearl    | 5x5de      | 0                         | 0                      |
> > > | pegs     | 4x4random  | 0                         | 0                      |
> > > | range    | 3x3        | 0                         | 0                      |
> > > | rect     | 3x2        | 0                         | 0                      |
> > > | samegame | 2x3c3s2    | 0                         | 40                     |
> > > | signpost | 2x3        | 0                         | 0                      |
> > > | singles  | 2x3de      | 0                         | 0                      |
> > > | sixteen  | 2x3        | 0                         | 0                      |
> > > | slant    | 2x2de      | 0                         | 0                      |
> > > | solo     | 2x2        | 0                         | 0                      |
> > > | tents    | 4x4de      | 0                         | 0                      |
> > > | towers   | 3de        | 0                         | 0                      |
> > > | tracks   | 4x4de      | 0                         | 0                      |
> > > | twiddle  | 2x3n2      | 0                         | 20                     |
> > > | undead   | 3x3de      | 0                         | 0                      |
> > > | unequal  | 3de        | 0                         | 0                      |
> > > | unruly   | 6x6dt      | 0                         | 0                      |
> > > | untangle | 4          | 0                         | 0                      |

---

### Official Review · Reviewer_oMdw · 2024-07-25
**a benchmark on algorithmic reasoning capabilities in RL**

**Rating:** 6
**Confidence:** 3
**Correctness:** I did not notice any major mistakes.
**Clarity:** Yes, it is clear.

**Review:**

See "Strengths" and "Opportunities For Improvement".

**Strengths:**

The paper has several strengths:

- PUZZLES provides a diverse set of 40 logic puzzles, offering a wide range of challenges for RL agents. This variety allows for a more thorough evaluation of algorithmic reasoning capabilities.

- PUZZLES addresses a significant gap in RL benchmarks, specifically targeting algorithmic and logical reasoning in single-agent, perfect-information environments.

- The ability to adjust puzzle sizes and difficulty levels is a significant strength, enabling researchers to test the generalization and adaptability of RL algorithms.

- The paper provides extensive evaluations of various RL algorithms, offering valuable insights into current capabilities and limitations.

- The demonstration of action masking's positive impact on training performance is a valuable contribution to RL research.

**Additional Feedback:**

N/A

**Documentation:**

Yes

**Limitations:**

The authors have discussed some limitations in the paper.

**Opportunities For Improvement:**

- While the paper mentions the potential of graph neural networks (GNNs), it doesn't explore these or other advanced architectures in depth.

- While the paper uses Simon Tatham's Puzzle Collection, there's limited discussion on why these specific puzzles were chosen and if they represent an optimal set for testing algorithmic reasoning.

- The baseline evaluation uses sparse rewards (only at puzzle completion), which may not be optimal for all puzzles or learning algorithms. While this is acknowledged, more exploration of alternative reward structures could be beneficial.

- The paper doesn't provide comparisons to human performance on these puzzles, which could offer valuable context for evaluating AI performance.

- While the paper identifies challenges, a deeper analysis of why certain algorithms fail on specific puzzles could provide more insights for future improvements.

**Relation To Prior Work:**

Yes

**Summary And Contributions:**

This paper introduces PUZZLES, a new benchmark for evaluating algorithmic reasoning capabilities in reinforcement learning (RL) agents.

PUZZLES is based on Simon Tatham's Portable Puzzle Collection, containing 40 diverse logic puzzles. The puzzles have adjustable sizes and difficulty levels, allowing evaluation of agent generalization. The benchmark provides both visual and discrete input options, with a discrete action space. The authors implemented PUZZLES as a Gymnasium environment for standardized RL research. And the benchmark allows testing of generalization capabilities across different puzzle sizes and difficulties.

Baseline evaluations were conducted using various RL algorithms, including PPO, DreamerV3, and MuZero. Results show that while some algorithms performed well on certain puzzles, many remain challenging, highlighting areas for improvement in RL. Experiments demonstrated the benefits of action masking and using discrete internal state observations over pixel-based inputs.

Overall, PUZZLES provides a comprehensive and flexible platform for evaluating and improving the logical reasoning capabilities of RL agents, with potential implications for advancing the field of machine learning.

---

> ### Author Rebuttal · Authors · 2024-08-16
>
> We thank the reviewer for their effort on reviewing our work. We address their concerns in the following.
>
> > While the paper mentions the potential of graph neural networks (GNNs), it doesn't explore these or other advanced architectures in depth.
>
> We believe that GNNs are an interesting direction for future research. However, we chose not to explore GNNs further at this stage for several reasons. First, defining the graph structure is non-trivial, with numerous design decisions that could reduce the benchmark's generalizability. Additionally, not all puzzles are well-suited for a graph-based approach. GNNs are also not included by default in any commonly used RL framework (e.g., Stable Baselines 3, RLlib). There is no standardized approach for applying GNNs to these tasks, making it challenging to establish a solid baseline for future work.
>
>
> > While the paper uses Simon Tatham's Puzzle Collection, there's limited discussion on why these specific puzzles were chosen and if they represent an optimal set for testing algorithmic reasoning.
>
> In order to provide a wide range of puzzles, we selected the largest single open-source collection of high-quality logic puzzles available and included all of those puzzles in our benchmark.
> The collection features popular puzzles like Sudoku, Mines, etc. While defining an optimal set is challenging, the diversity of puzzles, their popularity among human players, and their inherent difficulty make this selection a great test set for algorithmic reasoning. Moreover, we offer detailed instructions on our GitHub for adding new puzzles, enabling further expansion of the benchmark.
>
> > The baseline evaluation uses sparse rewards (only at puzzle completion), which may not be optimal for all puzzles or learning algorithms. While this is acknowledged, more exploration of alternative reward structures could be beneficial.
>
> We agree that sparse rewards introduce an additional layer of difficulty. However, our primary objective is to offer baseline evaluations that are broadly applicable. Introducing intermediate rewards involves numerous (puzzle-dependent) design choices that could reduce the generalizability of the reward structure. Therefore, we chose to conduct all baseline evaluations using a unified, sparse reward structure, and we suggest that exploring intermediate rewards should be the focus of future research.
>
> > The paper doesn't provide comparisons to human performance on these puzzles, which could offer valuable context for evaluating AI performance.
>
> We are currently evaluating the performance of a human expert on all 40 puzzles, we will update our manuscript and this thread with the results accordingly.
>
> > While the paper identifies challenges, a deeper analysis of why certain algorithms fail on specific puzzles could provide more insights for future improvements.
>
> Sparse feedback poses significant challenges, suggesting that more intermediate rewards are necessary for stable training progress. Additionally, there is considerable variance among random seeds. Generalization experiments indicate that the underlying rules are not fully learned, highlighting the potential for more aligned architectures, such as the previously mentioned GNNs. It is also interesting to note that strategies which need fewer steps in the training environment don’t generalize well to the generalization environment. Given the current low overall performance, it is premature to focus too much on specific puzzles. We will incorporate these insights into the revised manuscript.

---

> > ### Comment · Reviewer_oMdw · 2024-08-26
> >
> > Thank you for your comments. I will maintain my positive rating.

---

> > > ### Author Response · Authors · 2024-08-31
> > >
> > > Thank you for your reply.
> > >
> > > > We are currently evaluating the performance of a human expert on all 40 puzzles, we will update our manuscript and this thread with the results accordingly.
> > >
> > > Please find the results of the human expert evaluation in the following table. The human expert is able to solve 100% of the puzzles, all within our estimate of the upper bound of steps required for an optimal strategy. Note that the puzzles were solved at what we define in the paper as the *easiest-for-humans* difficulty level. At this level of difficulty, the AI agents were only able to consistently solve the untangle and cube puzzles (see appendix E.5). Interestingly, the solution for cube found by DreamerV3 requires less steps than the human expert.
> > > As of now, we are missing results for 4 puzzles. We will add all results to the final version of the manuscript.
> > >
> > >
> > > | Puzzle      | Config   | Steps    |
> > > | ----------- | -------- | -------- |
> > > | blackbox    | w5h5m3M3 | 120      |
> > > | bridges     | 7x7m2    | 49,3     |
> > > | cube        | 4x4      | 31       |
> > > | dominosa    | 3        | 51,5     |
> > > | filling     | 9x7      | 123      |
> > > | flood       | 12x12    | 63       |
> > > | galaxies    | 7x7      | 244,5    |
> > > | guess       | c6p4g10Bm| 63       |
> > > | inertia     | 10x8     | 33,5     |
> > > | keen        | 4        | 89       |
> > > | lightup     | 7x7      | 115,6    |
> > > | loopy       | 7x7t0    | 292      |
> > > | magnets     | 6x5      | 103      |
> > > | map         | 20x15n30 | 149      |
> > > | mines       | 9x9n10   | 152      |
> > > | mosaic      | 3x3      | 43       |
> > > | net         | 5x5      | 125      |
> > > | netslide    | 3x3b1    | 40,5     |
> > > | palisade    | 5x5n5    | 147      |
> > > | pattern     | 10x10    | 660      |
> > > | pearl       | 6x6de    | 245      |
> > > | range       | 9x6      | 305,5    |
> > > | rect        | 7x7      | 168,5    |
> > > | samegame    | 5x5c3s2  | 37       |
> > > | signpost    | 4x4      | 127      |
> > > | singles     | 5x5de    | 29,5     |
> > > | sixteen     | 3x3      | 88,5     |
> > > | slant       | 5x5de    | 162,3    |
> > > | solo        | 2x2      | 58,5     |
> > > | tents       | 8x8de    | 262      |
> > > | towers      | 4de      | 104      |
> > > | tracks      | 8x8de    | 430      |
> > > | undead      | 4x4de    | 68,5     |
> > > | unequal     | 4de      | 85       |
> > > | unruly      | 8x8dt    | 375,5    |
> > > | untangle    | 6        | 30,5     |

---

### Official Review · Reviewer_3eo1 · 2024-07-26
**PUZZLES: A Benchmark for Neural Algorithmic Reasoning**

**Rating:** 5
**Confidence:** 5

**Review:**

The paper presents a well-structured and comprehensive evaluation of the proposed benchmark. The inclusion of baseline algorithms and detailed experimental results strengthens the paper's quality. The authors have made a clear effort to provide sufficient details about the environment and experimental setup.

**Strengths:**

The PUZZLES benchmark has the potential to be a valuable tool for advancing research in algorithmic reasoning and reinforcement learning. By providing a standardized platform for evaluating agents on a diverse set of logic puzzles, the paper contributes to the field. However, the significance of the benchmark would be further strengthened by a more extensive evaluation of different agent architectures and a deeper analysis of the challenges posed by the puzzles.

Comprehensive evaluation of multiple RL algorithms on the benchmark
Clear explanation of the puzzle environment and its features
Potential to assess generalization capabilities of RL agents
Contribution to the development of algorithmic reasoning in RL

**Additional Feedback:**

Benchmark Expansion
Increase puzzle diversity: Incorporate puzzles from different domains (e.g., mathematical, logical, spatial) to broaden the scope of the benchmark.
Dynamic puzzle generation: Explore methods for generating new puzzle instances automatically, allowing for infinite variation and increased difficulty.
Multi-step puzzles: Introduce puzzles that require multiple intermediate steps or subgoals to solve, increasing the complexity of the task.
Evaluation Methodology
Human-agent comparison: Establish a human performance baseline for each puzzle to provide a reference point for agent performance.
Error analysis: Conduct a detailed analysis of agent errors to identify common failure modes and inform improvements.
Transfer learning evaluation: Assess the ability of agents to transfer knowledge between different puzzles or puzzle types.
Benchmark Accessibility
Open-source release: Make the benchmark code and data publicly available to foster collaboration and reproducibility.
Standardized evaluation framework: Provide a standardized evaluation framework to facilitate comparisons between different agents and algorithms.

**Clarity:**

The paper is generally clear and well-written, with a logical flow of ideas. The technical details are presented in a comprehensible manner, making it accessible to both experts and non-experts in the field. However, some additional visualizations or explanations could be beneficial for certain sections, such as the grid environment or the puzzle complexity.

**Correctness:**

While the concept of using logic puzzles as a benchmark is not entirely novel, the specific design choices of PUZZLES, such as the focus on algorithmic reasoning and the inclusion of various difficulty levels, contribute to the originality of the work. The paper effectively highlights the differences between PUZZLES and existing benchmarks.

**Documentation:**

Sentence structure: Some sentences could be restructured for better clarity and readability.
Visualizations: Additional figures or visualizations could be used to enhance the understanding of certain concepts or results.
Discussion of limitations: A more in-depth discussion of the benchmark's limitations and potential biases would strengthen the paper.

**Ethics:**

The work primarily involves developing a benchmark for evaluating AI agents on logic puzzles, which seems to be ethically neutral.

**Limitations:**

Benchmark Scope
Focus on logic puzzles: While the benchmark covers a diverse set of logic puzzles, it might not fully capture the breadth of algorithmic reasoning challenges. Other types of puzzles or algorithmic tasks could provide additional insights.
Limited puzzle complexity: While the benchmark offers adjustable difficulty levels, the current set of puzzles might not be sufficiently challenging for state-of-the-art RL agents. More complex and computationally demanding puzzles could be incorporated.
Evaluation Methodology
Limited evaluation metrics: The paper primarily focuses on the number of steps required to solve a puzzle, which might not fully capture the complexity of the problem or the quality of the learned solutions. Additional metrics could be considered, such as solution accuracy or generalization performance.
Lack of human performance baseline: A comparison with human performance on the same puzzles would provide a valuable reference point for evaluating agent capabilities.
Generalization and Transfer Learning
Limited generalization testing: While the paper mentions some preliminary experiments on generalization, a more extensive evaluation of how agents generalize to different puzzle sizes and types would be beneficial.
Lack of transfer learning experiments: Investigating the ability of agents to transfer knowledge learned on one puzzle to another could provide insights into the underlying learning mechanisms.

**Opportunities For Improvement:**

Limited experimental results and analysis, particularly regarding the impact of different puzzle types on agent performance
Lack of detailed discussion on the selection of puzzles and their representative nature
Potential limitations of the grid-based representation for certain types of puzzles

**Relation To Prior Work:**

It clearly outlines the limitations of existing benchmarks, such as their focus on game playing or motor control, and highlights the unique focus of PUZZLES on logical and algorithmic reasoning.

However, the paper could be strengthened by providing a more in-depth comparison of PUZZLES with other relevant benchmarks in the field of algorithmic reasoning, such as those mentioned in the related work section. This would allow for a clearer articulation of the specific novelties and contributions of PUZZLES.

Additionally, a more detailed discussion of how the puzzle selection process was conducted and the criteria used to determine the inclusion of specific puzzles would enhance the paper's clarity and originality.

**Summary And Contributions:**

This paper proposes PUZZLES, a new benchmark environment designed to evaluate an agent's ability to perform logical and algorithmic reasoning within the context of reinforcement learning (RL).

Here are the key points:

Motivation: Current RL methods excel at tasks like game playing and control, but struggle with tasks requiring logical and algorithmic reasoning.
PUZZLES Environment: This benchmark consists of 40 diverse logic puzzles with adjustable difficulty levels. These puzzles require the agent to discover an algorithm to solve them, not just through trial and error.
Features:
Supports visual or discrete input and discrete action space.
Difficulty can be scaled by adjusting puzzle size and complexity.
Allows for user-defined reward structures.
Benefits:
Provides a standardized platform for evaluating RL methods in logical and algorithmic reasoning tasks.
Enables researchers to assess an agent's ability to generalize to unseen puzzle configurations.
Evaluation: The paper evaluates various RL algorithms on PUZZLES. While some algorithms achieve reasonable success, there is still room for improvement, indicating the challenge PUZZLES presents.
Future Work: The authors suggest using the easiest human difficulty level as a baseline for future evaluations. Additionally, they propose exploring advanced neural network architectures like graph neural networks (GNNs) or Transformers to potentially improve performance.

---

> ### Author Rebuttal · Authors · 2024-08-16
>
> We thank the reviewer for their effort on reviewing our work. We address their concerns in the following.
>
> > Limited experimental results and analysis, particularly regarding the impact of different puzzle types on agent performance.
>
> We have extensive experimental results and analyses in the appendix, which did not make it into the main text due to space constraints.
>
> > Lack of detailed discussion on the selection of puzzles and their representative nature
>
> In order to provide a wide range of puzzles, we selected the largest single open-source collection of high-quality logic puzzles available (Simon Tatham’s Portable Puzzle Collection) and included all of those puzzles in our benchmark.
> The collection features popular puzzles like Sudoku, Mines, etc. While defining an optimal set is challenging, the diversity of puzzles, their popularity among human players, and their inherent difficulty make this selection a great test set for algorithmic reasoning. Moreover, we offer detailed instructions on our GitHub for adding new puzzles, enabling further expansion of the benchmark.
>
> > Potential limitations of the grid-based representation for certain types of puzzles
>
> Such representations implicitly occur in many algorithmic real-world settings, therefore we believe this is not a limitation. Additionally, if an agent acts on the pixel representation of a puzzle, there are no constraints as to how the puzzle is implicitly represented.
>
> > Benchmark Scope Focus on logic puzzles: While the benchmark covers a diverse set of logic puzzles, it might not fully capture the breadth of algorithmic reasoning challenges. Other types of puzzles or algorithmic tasks could provide additional insights.
>
> On Github, we provide instructions on how to add additional puzzles to the benchmark. With the current range of 40 diverse puzzles, we cover a wide range of algorithmic tasks.
>
> > Limited puzzle complexity: While the benchmark offers adjustable difficulty levels, the current set of puzzles might not be sufficiently challenging for state-of-the-art RL agents. More complex and computationally demanding puzzles could be incorporated.
>
> On the contrary, our results indicate that the puzzles are in fact challenging for current RL agents, even when using easy difficulty settings. Some puzzles could not be solved at all (e.g. Loopy and Pearl). Only a small subset could be solved at human difficulty levels.
>
> > Evaluation Methodology Limited evaluation metrics: The paper primarily focuses on the number of steps required to solve a puzzle, which might not fully capture the complexity of the problem or the quality of the learned solutions. Additional metrics could be considered, such as solution accuracy or generalization performance.
>
> We report solution accuracy in Tables 1, 2 and 3 in the main text as well as in the detailed result Tables 9, 10 and 11 in the appendix. We evaluate generalization performance in Table 3 for a transformer-based agent. However, default architectures are not invariant to input size and, therefore, are unable to generalize to different puzzle sizes.
>
> > Lack of human performance baseline: A comparison with human performance on the same puzzles would provide a valuable reference point for evaluating agent capabilities.
>
> We are currently evaluating the performance of a human expert on all 40 puzzles, we will update our manuscript and this thread with the results accordingly.
>
> > Generalization and Transfer Learning Limited generalization testing: While the paper mentions some preliminary experiments on generalization, a more extensive evaluation of how agents generalize to different puzzle sizes and types would be beneficial.
>
> An extensive evaluation of generalization requires the implementation of novel architectures and the careful evaluation of many design choices. As a benchmark paper, our goal is to provide baselines of currently existing architectures.

---

> > ### Author Rebuttal · Authors · 2024-08-16
> >
> > > Lack of transfer learning experiments: Investigating the ability of agents to transfer knowledge learned on one puzzle to another could provide insights into the underlying learning mechanisms.
> >
> > This is certainly an interesting direction for future research. However, with the current state we argue that it is more beneficial to reach reasonable performance on single puzzles and generalization with respect to size and difficulty of a given puzzle before attempting to generalize to other puzzles.
> >
> >
> > > Sentence structure: Some sentences could be restructured for better clarity and readability.
> >
> > We would gladly improve the sentence structure to further enhance readability and understandability. Could the reviewer please point out some examples of this?
> >
> > > Visualizations: Additional figures or visualizations could be used to enhance the understanding of certain concepts or results.
> >
> > We would gladly improve the manuscript to further enhance understandability. Could the reviewer please point out some examples of this?
> >
> > > Dynamic puzzle generation: Explore methods for generating new puzzle instances automatically, allowing for infinite variation and increased difficulty.
> >
> > Our codebase already provides this feature. Every puzzle instance is dynamically generated during runtime based on a random seed and a specification for size and difficulty. This allows for nearly infinite variations and adjustable difficulty.
> >
> > > Multi-step puzzles: Introduce puzzles that require multiple intermediate steps or subgoals to solve, increasing the complexity of the task.
> >
> > Our codebase already provides this feature. Many of the provided puzzles require multiple steps to be solved.
> >
> > > Benchmark Accessibility Open-source release: Make the benchmark code and data publicly available to foster collaboration and reproducibility.
> >
> > We have open-sourced the codebase under a permissive license on Github.
> >
> > > Standardized evaluation framework: Provide a standardized evaluation framework to facilitate comparisons between different agents and algorithms.
> >
> > We have added additional details for evaluation in the codebase on Github.

---

> > > ### Author Response · Authors · 2024-08-31
> > >
> > > > We are currently evaluating the performance of a human expert on all 40 puzzles, we will update our manuscript and this thread with the results accordingly.
> > >
> > > Please find the results of the human expert evaluation in the following table. The human expert is able to solve 100% of the puzzles, all within our estimate of the upper bound of steps required for an optimal strategy. Note that the puzzles were solved at what we define in the paper as the *easiest-for-humans* difficulty level. At this level of difficulty, the AI agents were only able to consistently solve the untangle and cube puzzles (see appendix E.5). Interestingly, the solution for cube found by DreamerV3 requires less steps than the human expert.
> > > As of now, we are missing results for 4 puzzles. We will add all results to the final version of the manuscript.
> > >
> > >
> > > | Puzzle      | Config   | Steps    |
> > > | ----------- | -------- | -------- |
> > > | blackbox    | w5h5m3M3 | 120      |
> > > | bridges     | 7x7m2    | 49,3     |
> > > | cube        | 4x4      | 31       |
> > > | dominosa    | 3        | 51,5     |
> > > | filling     | 9x7      | 123      |
> > > | flood       | 12x12    | 63       |
> > > | galaxies    | 7x7      | 244,5    |
> > > | guess       | c6p4g10Bm| 63       |
> > > | inertia     | 10x8     | 33,5     |
> > > | keen        | 4        | 89       |
> > > | lightup     | 7x7      | 115,6    |
> > > | loopy       | 7x7t0    | 292      |
> > > | magnets     | 6x5      | 103      |
> > > | map         | 20x15n30 | 149      |
> > > | mines       | 9x9n10   | 152      |
> > > | mosaic      | 3x3      | 43       |
> > > | net         | 5x5      | 125      |
> > > | netslide    | 3x3b1    | 40,5     |
> > > | palisade    | 5x5n5    | 147      |
> > > | pattern     | 10x10    | 660      |
> > > | pearl       | 6x6de    | 245      |
> > > | range       | 9x6      | 305,5    |
> > > | rect        | 7x7      | 168,5    |
> > > | samegame    | 5x5c3s2  | 37       |
> > > | signpost    | 4x4      | 127      |
> > > | singles     | 5x5de    | 29,5     |
> > > | sixteen     | 3x3      | 88,5     |
> > > | slant       | 5x5de    | 162,3    |
> > > | solo        | 2x2      | 58,5     |
> > > | tents       | 8x8de    | 262      |
> > > | towers      | 4de      | 104      |
> > > | tracks      | 8x8de    | 430      |
> > > | undead      | 4x4de    | 68,5     |
> > > | unequal     | 4de      | 85       |
> > > | unruly      | 8x8dt    | 375,5    |
> > > | untangle    | 6        | 30,5     |

---

### Decision · Program_Chairs · 2024-09-26

**Decision:**

Accept (Poster)

**Comment:**

This paper introduces PUZZLES, a benchmark for evaluating algorithmic reasoning in RL agents using 40 diverse puzzles from Simon Tatham's collection. The benchmark allows for adjustable puzzle complexity and tests generalization, with baseline results across a variety of RL algorithms. Reviewers noted challenges in areas such as generalization, transfer learning, sparse rewards, and result variance, but the authors addressed these concerns through reasonable updates, including human expert and LLM evaluations. Given its potential impact and the authors' constructive responses during the discussion, I recommend acceptance. PUZZLES offers a valuable contribution to RL and potentially LLM research in algorithmic reasoning.